*Resource*

# Decreased blood vessel density and endothelial cell subset dynamics during ageing of the endocrine system

Junyu Chen[1,2], Luciana Lippo[1], Rossella Labella[1], Sin Lih Tan[1], Brian D Marsden[3,4] (iD),
Michael L Dustin[3] (iD), Saravana K Ramasamy[5,6] & Anjali P Kusumbe[1,*] (iD)

## Abstract

Age-associated alterations of the hormone-secreting endocrine system cause organ dysfunction and disease states. However, the cell biology of endocrine tissue ageing remains poorly understood. Here, we perform comparative 3D imaging to understand age-related perturbations of the endothelial cell (EC) compartment in endocrine glands. Datasets of a wide range of markers highlight a decline in capillary and artery numbers, but not of perivascular cells in pancreas, testis and thyroid gland, with age in mice and humans. Further, angiogenesis and β-cell expansion in the pancreas are coupled by a distinct age-dependent subset of ECs. While this EC subpopulation supports pancreatic β cells, it declines during ageing concomitant with increased expression of the gap junction protein Gja1. EC-specific ablation of Gja1 restores β-cell expansion in the aged pancreas. These results provide a proof of concept for understanding age-related vascular changes and imply that therapeutic targeting of blood vessels may restore aged endocrine tissue function. This comprehensive data atlas offers over > 1,000 multicolour volumes for exploration and research in endocrinology, ageing, matrix and vascular biology.

**Keywords** 3D imaging; ageing; endocrine system; pancreas; vasculature
**Subject Categories** Cell Adhesion, Polarity & Cytoskeleton; Methods & Resources; Vascular Biology & Angiogenesis
**The EMBO Journal (2021) 40: e105242**

## Introduction

The vascular network functions as a critical regulator of vertebrate physiology through its well-established function of nutrient and oxygen transport. In addition, blood vessels provide factors termed angiocrine signals (Butler *et al*, 2010; Ding *et al*, 2014; Rafii *et al*, 2015; Singh *et al*, 2019) that control organ development and stem cell behaviour (Butler *et al*, 2010; Ding *et al*, 2012; Tashiro *et al*, 2012; Rafii *et al*, 2016; Ramasamy *et al*, 2016; Wei & Frenette, 2018). Tissue-specific capillary beds support a distinctive function of each organ and respond to dynamically changing local needs (Nolan *et al*, 2013; Rafii *et al*, 2016; Augustin & Koh, 2017). This is particularly true for the endocrine system, which provides spatial, rapid and dynamic hormone output in response to systemic and regional changes (Tsang *et al*, 2016; Nadal *et al*, 2017; Russell & Lightman, 2019). Endocrine glands are highly vascularized (Katoh, 2003; Gorczyca *et al*, 2010), and vasculature is suggested to control and facilitate this instant and dynamic hormone output (Ballian & Brunicardi, 2007; Jang *et al*, 2017; Le Tissier *et al*, 2017). Although recent studies on bone and liver vascular beds indicate organ-based specialization of the vasculature (Ding *et al*, 2010; Ding *et al*, 2014; Kusumbe *et al*, 2014; Ramasamy *et al*, 2014; Kusumbe, 2016; Coutu *et al*, 2017; Romeo *et al*, 2019), the structure and functions of vasculature in several other organ systems, including the endocrine glands, remain poorly studied. The physiology of the endocrine system changes substantially throughout life, and ageing impacts hormone production and endocrine function (Smith & Xu, 2008; Veldhuis, 2013; Vitale *et al*, 2013; Chaker *et al*, 2018). However, a systematic understanding of age-dependent changes in tissue microenvironments of endocrine glands remains elusive.

Recent studies indicate that vascular ageing leads to loss of stem and progenitor cell functions in the skeletal system (Asumda & Chase, 2011; Kusumbe *et al*, 2016). Identifying age-related vascular changes to elucidate mechanisms driving ageing holds potential to manipulate blood vessels to rejuvenate the aged endocrine tissues (Almaça *et al*, 2014). Such observations favour investigation of communication between ECs with other cell types in the endocrine tissues. For instance, in the pancreas of the endocrine system, active

1 Tissue and Tumor Microenvironments Group, The Kennedy Institute of Rheumatology, University of Oxford, Oxford, UK
2 Department of Prosthodontics, State Key Laboratory of Oral Diseases, West China Hospital of Stomatology, Sichuan University, Chengdu, China
3 The Kennedy Institute of Rheumatology, University of Oxford, Oxford, UK
4 Structural Genomics Consortium, NDM, University of Oxford, Oxford, UK
5 Institute of Clinical Sciences, Imperial College London, London, UK
6 MRC London Institute of Medical Sciences, Imperial College London, London, UK
*Corresponding author. Tel: +44 01865 612 688; E-mail: anjali.kusumbe@kennedy.ox.ac.uk

pancreatic β-cell replication declines upon ageing (Chen et al, 2011; Kushner, 2013). Acquired tissue insufficiency, which is more pronounced in ageing, underlies the pathogenesis of diverse human diseases; for example, absolute or relative deficits in pancreatic β cells underlie diabetes mellitus (Butler et al, 2003; Donath & Halban, 2004; Meier & Bonadonna, 2013). Identifying the factors for this decline may reveal strategies for inducing β-cell expansion, a long-sought goal for diabetes therapy (Vetere et al, 2014).Similarly, a decline in tissue architecture and testosterone production of the testes in ageing negatively impacts fertility (Harris et al, 2011).

Here, in this study, we generate cell–cell interactome and 3D spatial proteomic data by performing deep imaging of whole glands at a single-cell resolution, which provides novel insights into age-dependent vascular changes. We identify a phenotypically and functionally distinct subset of capillaries which maintain β cells in pancreatic islets. These capillaries decline with age, and their reactivation led to the restoration of β-cell self-renewal and numbers (Appendix Fig S1). Thus, β cells and associated blood vessels form a functional unit in the pancreas with a critical role in insulin production. Similarly, enhanced angiogenesis boosted testosterone production in mice. Our findings provide proof of concept for manipulation and restoration of blood vessels in aged endocrine tissues to rejuvenate the endocrine tissues.

## Results

### 3D atlas of vascular cells and molecules in young and aged endocrine glands

To understand vascular microenvironments and their age-related changes in endocrine glands, we screened ~ 150 antibodies for mapping EC, pericyte, stromal cell and matrix to understand tissue distribution in selected specimens. We then selected ~ 50 of these antibodies that were most informative in defining vascular microenvironments and extended the analysis to 2,700 samples. We provide detailed information on the working conditions of these antibodies and also exemplar images of the negative controls (Appendix Table S1 and Appendix Fig S2A). We performed deep imaging followed by computational surface rendering to generate datasets of whole young (2–8 weeks old) and aged (56–70 weeks old) mouse adrenal gland, ovary, pituitary gland, testis, pancreas and thyroid gland (Fig 1A). Thick longitudinal cryosections spanning the endocrine glands were generated and subjected to multiplex immunolabelling by the pairing of primary antibodies with optimal fluorophore-tagged secondary antibody combinations. Tissue sections were imaged to generate three-dimensional high magnification images on a laser scanning confocal microscope at a single-cell resolution. The images were stitched to generate whole-gland single-cell resolution 3D maps (Fig 1A) and subsequently subjected to computational surface rendering for the analysis of interactions between different cell types (Fig 1B). The exemplar images show that the organization, expression patterns and spatial distributions of ECs and other cell types such as pericytes and stromal cells are evident in our 3D volumes (Fig 1A and Movies EV1–EV11) enabling the generation of cellular cartography for these cell types across the endocrine system. Taking the example of the adrenal gland; the cortex was highly vascularized consisting of

column-like VCAM-1 negative radially arranged capillaries while the medulla harboured thick highly branched capillaries with a high expression of VCAM-1 (Fig 1A). The single-cell resolution nature of multicolour images permits the analysis of interactions between different cell types (Fig 1B) which provides a wealth of 3D spatial proteomic data and cell–cell interactomes based on the distance analysis of different cell markers (Fig 1C–E, Appendix Fig S2B and C). In the case of the adrenal gland, ECs of the capillaries in cortex, medulla and the transition region were distinguishable by marker expressions. Specifically, cortex ECs were $CD102^{pos}$/$VCAM\text{-}1^{lo}$/$Vimentin^{neg}$, ECs in the transition zone were $CD102^{neg}$/$VCAM\text{-}1^{hi}$/$Vimentin^{neg}$ and ECs in the medulla region were $CD102^{neg}$/$VCAM\text{-}1^{hi}$/$Vimentin^{pos}$ (Fig 1D). Moreover, these spatially and phenotypically distinct ECs were juxtaposed to different cell types indicating that they also had distinct cellular interactomes (Fig 1B–D). Further, we also extensively compared young and aged endocrine glands, so the database provides opportunities to mine age-related changes in cellular interactions. Following on the adrenal gland, the EC interactome in the adrenal gland demonstrated numerous changes upon ageing (Fig 1D). Thus, this database could serve as an essential research tool to understand tissue biology in various fields of physiology, ageing, matrix and vascular biology to investigate functional pathophysiology and therapeutic effects. Therefore, we make this extensive single-cell resolution 3D volumes with over 1000 three-dimensional scans of whole mouse endocrine glands publicly available for further exploration and quantitative analyses (Fig 1E and Table EV1). Maximum intensity projections of numerous of these high-resolution 3D tile scans of endocrine glands are provided in the expanded view data (Appendix Table S2–S12).

### Ageing is associated with a decline in capillary and artery numbers in mice and human endocrine glands

Investigation of vascular changes in ageing endocrine glands revealed endocrine tissue-specific features in young versus aged endocrine glands (Tables EV2 and EV3). We performed a comprehensive analysis for regional differences such as cortex versus medulla and periphery versus centre, blood vessel distributions, blood vessel density, capillary diameters, artery diameters and artery numbers (Figs 2 and 3 and Appendix Fig S3). In addition to the confocal imaging of thick sections, we also performed light-sheet imaging of the cleared whole-endocrine glands. Based on the 3D confocal images, adrenal glands and ovaries exhibited no significant change in microvascular density numbers, while the pituitary glands demonstrated an increase in microvascular density upon ageing (Fig 2A–C). The testis, thyroid gland and pancreas with numerous EC markers demonstrated the most remarkable age-related decline in microvascular density (Fig 2D–F). The light-sheet images of the cleared whole-endocrine glands confirmed the age-dependent changes in microvascular density (Fig 3A–E). Adrenal glands and ovaries showed no apparent differences in artery numbers (Fig 3A–C), while aged murine pancreas, thyroid gland and testis demonstrated a decline in artery numbers (Fig 3C–E). Moreover, all these age-related changes in microvascular density and arterial numbers in testis, pancreas and thyroid gland were validated by comparing the adult and aged mice (Appendix Fig S4). In line with the decline of blood vessels, aged pancreas, thyroid gland and testis showed abundant hypoxic regions as determined by the pimonidazole

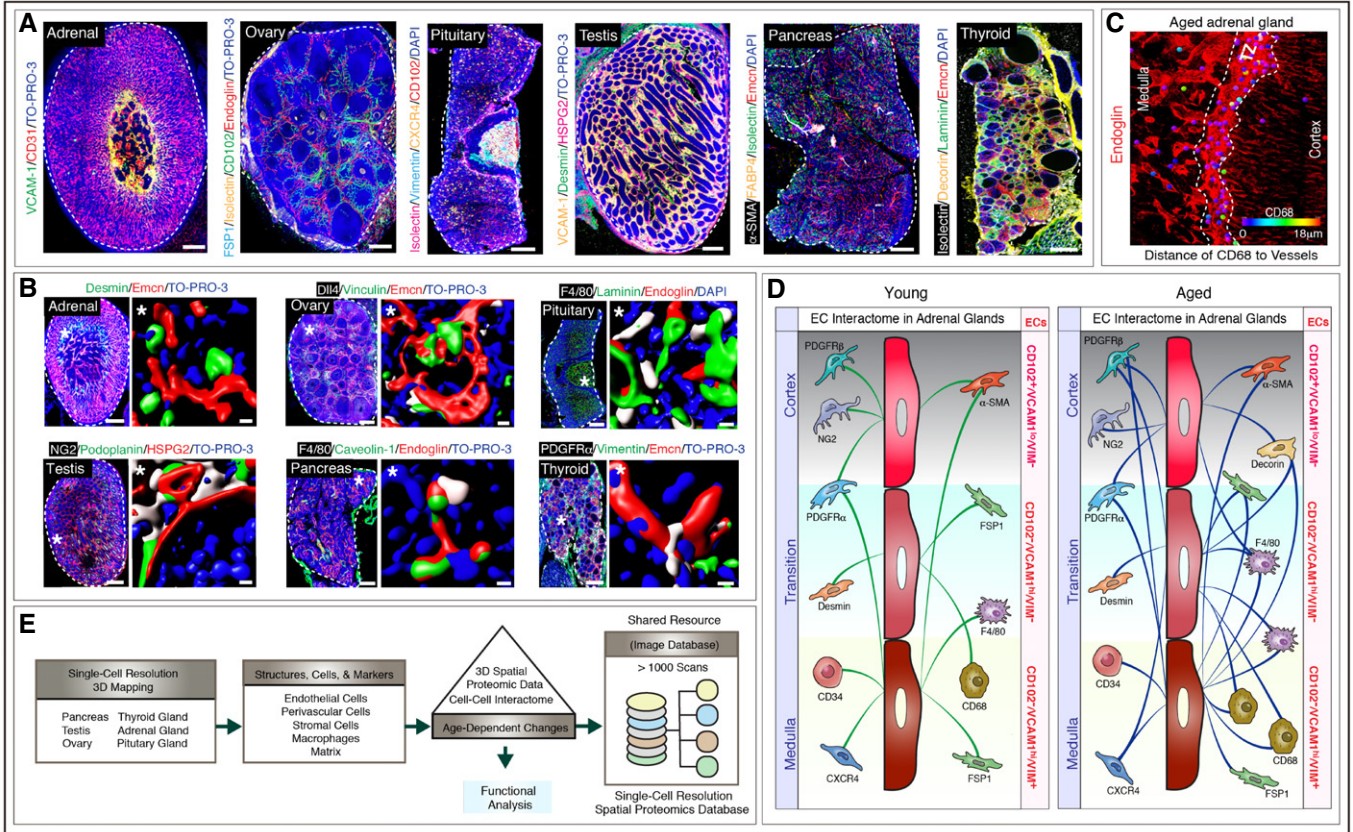

**Figure 1.  3D atlas of vascular-related cells and molecules in young and aged endocrine glands.**

A   Representative 3D tile scan images of young adrenal gland, ovary, pituitary gland, testis, pancreas and thyroid gland stained with the antibodies indicated in the panel. Scale bars are 200 μm. The white and dashed lines represent the outlines of each organ.

B   Representative 3D images at single-cell resolution level of adrenal gland, ovary, pituitary gland, testis, pancreas and thyroid gland. Insets (*) indicate higher magnification images of regions with single-cell resolution and surface rendering for all markers. Scale bars for tile scan 3D images are 200 μm; for single-cell resolution, insets are 5 μm. The dashed lines represent the outlines of each organ.

C   Exemplar image illustrates the cell–cell interactome analysis with the distance estimates of CD68$^+$ macrophages to blood vessels (Endoglin) in cortex, medulla and the transition zone (TZ) of an aged adrenal gland. The colour code indicates the distance which is very close in violet and very far in red. The scale bar is 50 μm. The white and dashed lines represent the outline of the transition zone.

D   Illustration of the EC interactome in adrenal glands and changes in the interactome with ageing. Cells ≤ 18 μm of distance from the blood vessels were considered to be within the interactome of blood vessels. Analysis of endothelium in the different region showed that they have distinct marker expressions. CD102$^{pos}$/VCAM-1$^{lo}$/ Vimentin$^{neg}$, CD102$^{neg}$/VCAM-1$^{hi}$/Vimentin$^{neg}$ and CD102$^{neg}$/VCAM-1$^{hi}$/Vimentin$^{pos}$ ECs were located in cortex, transition and medulla regions, respectively, in young and aged adrenal glands. Their interactions in young and aged adrenal glands with PDGFRβ$^+$ and NG2$^+$ pericytes, PDGFRα$^+$ mesenchymal cells, α-SMA$^+$ pericytes, Desmin$^+$ stromal cells, Decorin$^+$ stromal cells, CXCR4$^+$ stromal cells, FSP1$^+$ fibroblasts, CD34$^+$ hematopoietic cells, F4/80$^+$ macrophages and CD68$^+$ macrophages are shown. Interaction with Decorin expressing stromal cells is only observed in ageing.

E   Schematic illustration of the study design with 3D spatial proteomic data and cell–cell interactome database strategy, analysis of age-dependent changes and validation of protein expression followed by functional analysis.

Data information: Nuclei: DAPI or TO-PRO-3 as indicated.

administration (Appendix Fig S5). Despite the hypoxic nature of aged glands, HIF1α was not stabilized or overexpressed (Appendix Fig S5). Notably, an increase in perivascular macrophages and fibroblasts was commonly observed in ageing endocrine glands (Appendix Fig S6). Although comparison of young and aged endocrine glands demonstrated tissue-specific expression patterns and age-associated differentially regulated candidates in 3D spatial proteomic data based on average expression intensity analysis (Fig 3F), pericyte numbers and their distribution remained unaltered (Fig EV1A–D).

To understand the relevance of age-related changes observed in murine endocrine glands, we analysed human endocrine tissues.

Healthy human tissues such as adrenal gland, ovary, testis, pancreas and thyroid gland of young (< 20 years) and aged (> 70 years) individuals (Table EV4) were analysed to understand vascular changes by performing immunohistochemistry and quantitative PCR (qPCR) for endothelial and perivascular cell markers. For instance, changes in arteries were determined by immunolabelling for α-SMA, a marker for artery-associated perivascular cells and transcript levels of *Efnb2*, a known marker for arterial ECs (Adams & Alitalo, 2007). Comparable to mouse endocrine tissues, aged human adrenal glands and ovaries showed no apparent differences in the microvascular density and artery numbers (Fig 4A and B). Moreover, similar to mice, aged human testis, pancreas and thyroid

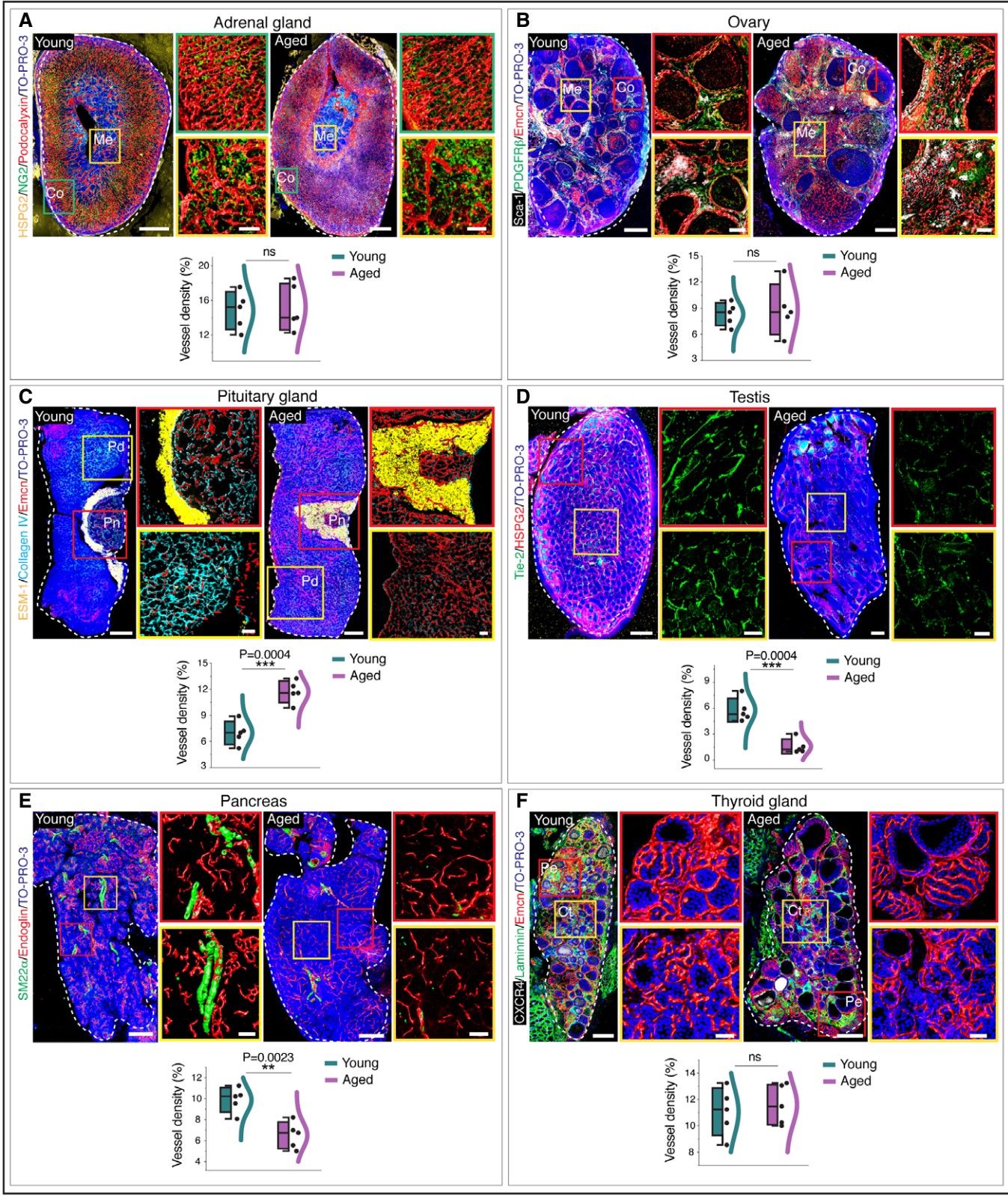

Figure 2.

◄

**Figure 2. Age-associated vascular modulations in mouse endocrine glands.**

A  Young and aged adrenal gland immunolabelled with HSPG2, NG2 and Podocalyxin. Higher magnification insets show cortex (Co) and medulla (Me). Combined box and whiskers, and scatter plot with quantification of vessel density in young and aged adrenal glands.
B  Young and aged ovary immunolabelled with Sca-1, PDGFRβ and Emcn. Higher magnification insets show cortex and medulla. Combo plot with quantification of vessel density in young and aged ovary.
C  Young (10 weeks old) and aged pituitary gland immunolabelled with ESM-1, collagen IV and Emcn. Insets with higher magnifications of pars distalis and pars nervosa regions. Combo plot shows quantification of vessel density.
D  Young and aged testis immunolabelled with Tie-2 and HSPG2. Higher magnifications of cortex and medulla. Combo plot shows quantification of vessel density.
E  Young and aged pancreas immunolabelled with SM22α and Endoglin. Insets representative of higher magnifications of regions in young and aged pancreas. Combo plot shows quantification of vessel density.
F  Young and aged thyroid gland immunolabelled with CXCR4, laminin and Emcn. Combo plot shows quantification of vessel density in young and aged thyroid glands. Insets: periphery and centre of the thyroid gland.

Data information: The white and dashed lines in each panel represent the outlines of each organ. ($n$ = 5), $P$-values and two-tailed unpaired $t$-tests. ns: not significant; **$P$ < 0.01; ***$P$ < 0.001. Nuclei: TO-PRO-3. For the combo plots, the boxes represent mean ± s.d., line in the box is the median, and the lower and upper lines show the minimum and the maximum of the values. The line on the right side of these combined plots represents the sample distribution. Co, cortex; Ct, centre; Me, medulla; Pd, pars distalis; Pe, periphery; Pn, pars nervosa. Scale bars for tile scan 3D images 200 μm, for insets 50 μm.

gland exhibited a decline in microvascular density and arterial numbers (Fig 4C–E). Importantly, analysis of even 4 random human samples of each of these glands showed these significant changes. Further, transcript analysis confirmed the age-related changes observed in human pancreas, testis and thyroid gland (Fig 4F). The abundance of fibroblasts (Appendix Fig S7) and unaltered pericyte numbers (Fig EV1E–I) in aged human glands similar to the murine endocrine glands confirmed similarity in the ageing of endocrine tissues between mouse and humans. Endocrine glands are highly vascularized, a feature critical for endocrine cells to rapidly sense fluctuations in blood composition to produce and secrete hormones (Katoh, 2003; Ballian & Brunicardi, 2007; Gorczyca *et al*, 2010; Jang *et al*, 2017; Le Tissier *et al*, 2017). Thus, the observed decline of blood vessels with age in endocrine tissues could influence the hormone output and thereby the optimal function of an endocrine gland.

## Identification of a subpopulation of age-dependent ECs in pancreatic endocrine tissue

Structural and phenotypic analysis of vasculature in endocrine tissues highlighted the existence of blood vessel heterogeneity in the pancreas. Consistent with previous reports, blood vessels within islets of adult pancreas showed a dense network of highly branched thick capillaries (Zhou *et al*, 1996; Gorczyca *et al*, 2010) (Fig EV2A and B) in comparison to the surrounding non-islet blood vessels which were typical tissue-associated quiescent and thin capillaries (Fig EV2A and B). At the interface of islets and exocrine tissues, these two types of capillaries were connected (Fig EV2A), indicating that they are a part of one continuous vascular bed. Interestingly, capillaries within pancreatic islets demonstrated a dramatic decline in abundance with ageing in comparison to capillaries in the exocrine tissue (Fig 5A). Notably, blood vessels in the mouse adult (19 weeks old) pancreatic islets were highly angiogenic with high numbers of proliferating ECs and harbouring cells expressing tip cell marker ESM-1 (Fig 5B and C, and EV2C). Islet capillaries were strongly positive for endomucin (Emcn), while capillaries in the exocrine region of pancreas displayed a low Emcn expression (Fig 5D). Combining a pan-endothelial marker CD31 (also known as Pecam1), the phenotypically distinct capillaries in pancreatic islets could be identified as CD31$^+$ Emcn$^{hi}$ ESM-1$^+$ blood vessels (Fig 5C and D). Thus, islet capillaries could be demarcated and isolated

using CD31$^+$ Emcn$^{hi}$ ESM-1$^+$ expression in ECs compared to CD31$^+$ Emcn$^{lo}$ cells in exocrine regions of the pancreas (Fig 5C–E). Analysis of arteries in relation to capillaries indicated that α-SMA$^+$ arteries and arterioles terminated at the CD31$^+$ Emcn$^{hi}$ ESM-1$^+$ islet capillaries (Fig EV2D).

During physiological growth in humans, mice and other species, juvenile β cells expand by self-renewal while maintaining their hallmark functions (Dor *et al*, 2004; Kushner *et al*, 2005). Pancreatic β-cell proliferation decreases with age (Chen *et al*, 2011; Kushner, 2013). In corroboration with these previous reports, our analysis demonstrated a decline in β-cell proliferation with age, first decline from juvenile to adult and a further decline in aged pancreas (Fig 5F and G). These changes correlated with a prominent decline of CD31$^+$ Emcn$^{hi}$ ESM-1$^+$ islet capillaries in ageing mice (Figs 5H and EV2E). Flow cytometry analysis confirmed this age-dependent decline of CD31$^+$ Emcn$^{hi}$ ECs (Fig 5E).

## Importance of angiogenic CD31$^+$ Emcn$^{hi}$ ESM-1$^+$ islet-specific ECs

VEGF-VEGFR2 signalling is a critical driver of physiological and pathological neo-angiogenesis, and loss of VEGFR2 on ECs inhibits angiogenesis (Abhinand *et al*, 2016; Simons *et al*, 2016; Zhang *et al*, 2019). Predominant angiogenesis in islet capillaries compared to the surrounding exocrine tissue capillaries prompted us to analyse VEGF distribution in adult pancreas. VEGFA, a potent angiogenic factor expression in adult pancreas (Brissova *et al*, 2006; Aamodt & Powers, 2017), showed the abundant distribution in islets (Figs 5I, and EV2F and G). The loss of CD31$^+$ Emcn$^{hi}$ capillaries and PLVAP$^+$ fenestrated vessels in aged pancreatic islets (Figs 5H and EV2H) correlated with the loss of VEGFA expression in the islets as determined by the immunostaining of young and aged tissues and ELISAs of isolated islets from 10- and 55-week-old mice (Fig 5I). We, therefore, targeted the VEGF signalling to understand the importance of islet capillaries in adult pancreas. Systemic administration of pharmacological drug sunitinib, a potent inhibitor of VEGFR2, demonstrated a decline in CD31$^+$ Emcn$^{hi}$ vessel density and ESM-1$^+$ cell numbers in adult mice (Figs 6A and B and EV2I). Reduction in islet capillaries in sunitinib-treated mice was associated with the reduction in insulin production, β-cell proliferation and numbers (Fig 6B–D). To study VEGF signalling specifically in islet ECs, we generated EC-specific loss-of-function mice (*Vegfr2*$^{iΔEC}$) by combining loxP-flanked *Vegfr2* alleles (*Vegfr2*$^{lox/lox}$) and *Cdh5(PAC)-CreERT2*

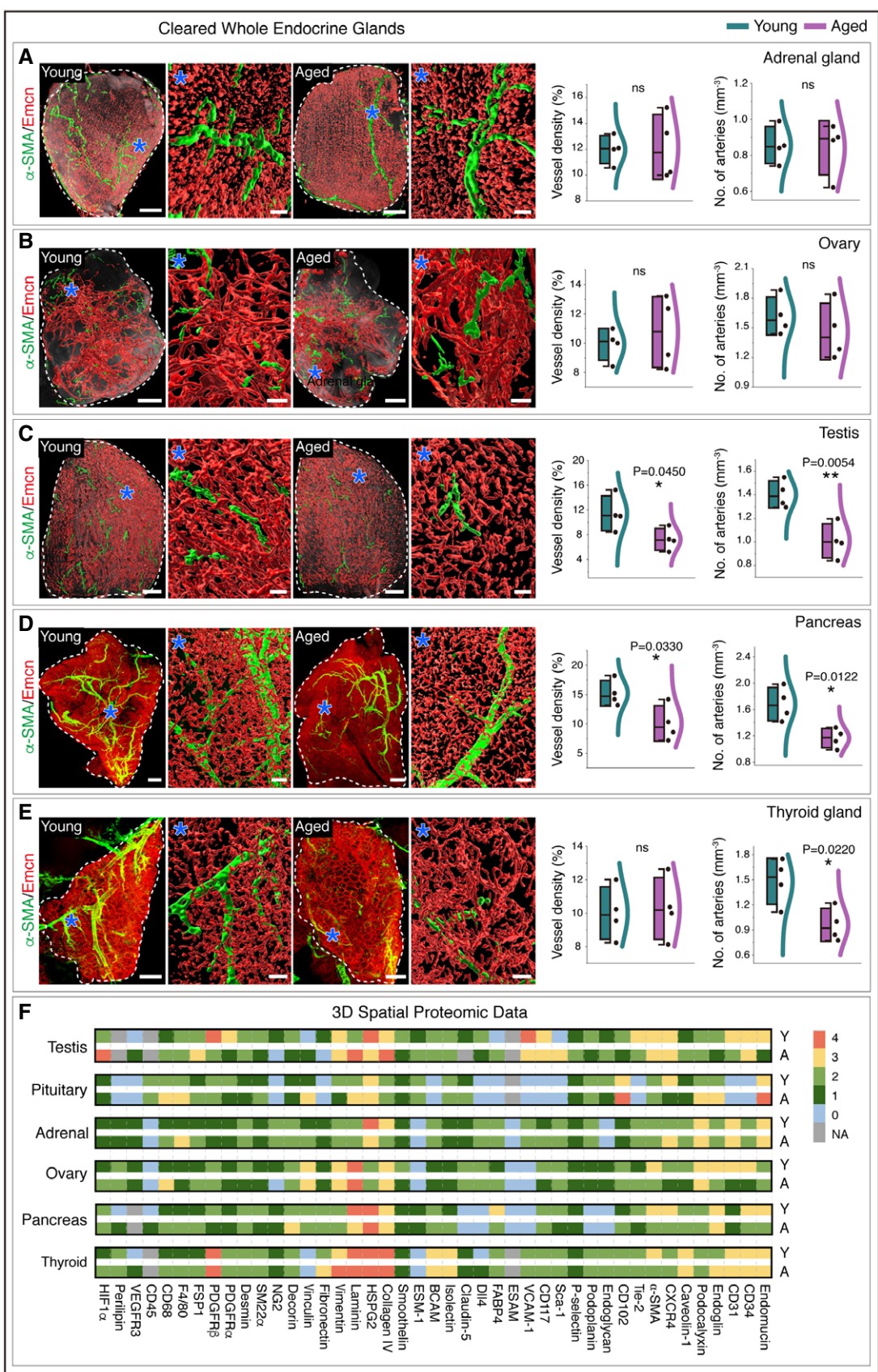

**Figure 3.**

**Figure 3. Light-sheet imaging of whole cleared endocrine glands from young and aged mice.**

A  Whole-endocrine gland imaging of young and aged adrenal glands immunolabelled with α-SMA and Emcn acquired on a light-sheet microscope. Asterisks indicate higher magnifications of regions in adrenal glands from young and aged mice. Combined box and whiskers, and scatter plots with quantifications of vessel density and numbers of arteries.

B  Whole-endocrine gland light-sheet imaging-based 3D images of young and aged ovaries stained with α-SMA and Emcn. Asterisks represent higher magnifications of regions in ovaries from young and aged mice. Combo plots show quantifications of vessel density and numbers of arteries in young and aged ovary.

C  3D images of whole cleared young and aged testes stained with α-SMA and Emcn. Asterisks indicate higher magnifications of indicated regions. Combo plots show quantifications of vessel density and numbers of arteries.

D  Light-sheet imaging of whole cleared young and aged pancreases immunolabelled with α-SMA and Emcn. Asterisks indicate higher magnifications of regions. Combo plots with quantifications of vessel density and numbers of arteries in young and aged pancreas.

E  3D images acquired on a light-sheet microscope of whole cleared young and aged thyroid glands stained with α-SMA and Emcn. Asterisks indicate higher magnifications of regions. Combo plots show quantifications of vessel density and numbers of arteries in young and aged thyroid glands.

F  Heatmap of 3D spatial proteomic data showing qualitative analysis of differences in cell type markers and matrix expressions between young and aged murine endocrine glands. The colour code based on immunolabelling intensities indicates very high expression in red, high expression in yellow, medium in light green, low in dark green and no expression in grey.

Data information: The white and dashed lines in panels (A–E) represent the outlines of each organ. ($n = 4$), $P$-values and two-tailed unpaired $t$-tests for all the above panels. ns, not significant; *$P < 0.05$; **$P < 0.01$. For the combo plots, the boxes represent mean ± s.d., line in the box is the median, and the lower and upper lines show the minimum and the maximum of the values. The line on the right side of the combo plots represents the sample distribution. Scale bars are 200 μm for whole-organ 3D images and 50 μm for higher magnification insets.

transgenics. Following tamoxifen administration in the adult 16-week-old mice and analysis of *Vegfr2*[iΔEC] pancreatic mutants at the age of 19 weeks revealed vascular defects within the islets (Fig 6E). In addition to perturbed angiogenesis, EC-specific *Vegfr2* loss of function led to decreases not only in CD31[+] Emcn[hi] vessel density and ESM-1[+] cell numbers (Figs 6F and EV2J) but also in β-cell numbers and β-cell proliferation (Fig 6G and H). Quantification of insulin production and islet mass showed a notable decrease in *Vegfr2*[iΔEC] mutants (Fig 6F). These findings suggested the importance of islet capillaries in regulating β cells in the pancreas. Further, the above observed loss of VEGFA within the islet/endocrine tissue upon ageing (Fig 5I) is also likely to impact the surrounding exocrine tissue blood vessel proliferation and maintenance.

To gain molecular insights, the expression of mRNAs for secreted growth factors with known roles in β-cell survival and proliferation (Kulkarni *et al*, 2002; Yano *et al*, 2007; Yesil & Lammert, 2008; Chen *et al*, 2011; Demirci *et al*, 2012; Alagpulinsa *et al*, 2019) was analysed in freshly purified CD31[+] Emcn[hi] ECs compared to CD31[+] Emcn[lo] ECs. *Pdgfa*, *Pdgfb*, *Igf1*, *Igf2*, *Cxcl12*, *Hgf* and *Kitl* transcripts were significantly higher expressed in CD31[+] Emcn[hi] ECs relative to CD31[+] Emcn[lo] ECs (Fig 6I). We also analysed transcript levels of these factors in human pancreas samples. Young human pancreatic tissues with high CD31[+] Emcn[hi] islet capillaries showed higher expression of *Pdgfa*, *Pdgfb*, *Igf1*, *Igf2*, *Cxcl12*, *Hgf* and *Kitl* compared to aged pancreas (Fig EV2K). Further, multiple of these growth factors are known to promote angiogenesis and are required to maintain blood vessels. The up-regulation of secreted factors by CD31[+] Emcn[hi] ECs and their decline in ageing including the loss of islet VEGFA with ageing (Fig 5I) indicate that the observed decline in blood vessels in the surrounding exocrine tissue of pancreas is mediated by islet/endocrine tissue.

Further assessment of the relationship between angiogenesis and β-cell expansion demonstrated that EC proliferation within islets positively correlated with β-cell proliferation (Fig 6J). These results showed that active angiogenesis and formation of CD31[+] Emcn[hi] vessels but not the quiescent vessels promoted β-cell expansion. Transcript analysis of factors involved in blood vessel growth and proliferation (Lu *et al*, 2004; Tammela *et al*, 2008; Herbert & Stainier, 2011; Kusumbe *et al*, 2014; Romeo *et al*, 2019) demonstrated

an up-regulation of *Dll4*, *Sele*, *Icam1*, *Plxnd1*, *Mmp9*, *Robo4*, *Unc5b*, *Kdr* and *Flt4* transcripts in CD31[+] Emcn[hi] subset (Fig 6K). In addition to growth factor signalling, cellular interactions between ECs drive angiogenic process (Carmeliet, 2003; Melgar-Lesmes & Edelman, 2015). Gene expression analysis showed an abundant increase in transcript levels of junctional proteins in islet ECs (Fig 6K). Notably, a gap junction protein Gja1, commonly known as Connexin 43, the most abundant isoform of gap junction protein on ECs, demonstrated a significant decrease in the CD31[+] Emcn[hi] subset (Fig 6K). Interestingly, *Gja1* showed an age-dependent expression in the CD31[+] Emcn[hi] EC subset with a significant increase from juvenile to adult and a further increase in aged pancreas (Fig 6L). Thus, the islet CD31[+] Emcn[hi] capillary EC subset exhibited specific expression profiles suggesting its specific functional properties.

**Endothelial Gja1 negatively regulates CD31[+] Emcn[hi] ESM-1[+] ECs**

Gja1 belongs to the family of transmembrane proteins that assemble as hexameric plasma membrane structures "connexons" (Solan & Lampe, 2009). Connexons function either as plasma membrane channel, termed "hemichannel", or dock head-to-head with another connexon from an adjacent cell self-assembling into a gap junction intercellular channel (Goodenough & Paul, 2009). To investigate the functional role of Gja1 in blood vessels of the endocrine system, inducible EC-specific loss-of-function mice (*Gja1*[iΔEC]) were generated by combining loxP-flanked *Gja1* alleles and *Cdh5(PAC)-CreERT2* transgenics. Following tamoxifen administration in adult mice and analysis at 19-week-old mice, the *Gja1*[iΔEC] mutant pancreatic islets displayed remarkable changes in the vasculature (Figs 7 and EV3). A profound increase in overall blood vessel density was observed in pancreas along with the expansion of CD31[+] Emcn[hi] islet capillaries in *Gja1*[iΔEC] mutants compared to the littermate control mice (Figs 7A–C, and EV3A and B). Artery numbers, proliferating CD31[+] Emcn[hi] EC numbers and VEGFA expression within the islets were increased in *Gja1*[iΔEC] mutant pancreas (Figs 7B and EV3C–E), revealing Gja1 as a negative regulator of blood vessel growth and EC proliferation in the pancreas. The changes in islet capillaries were accompanied by an increase in β-cell mass, β-cell proliferation and insulin production (Fig 7D–J).

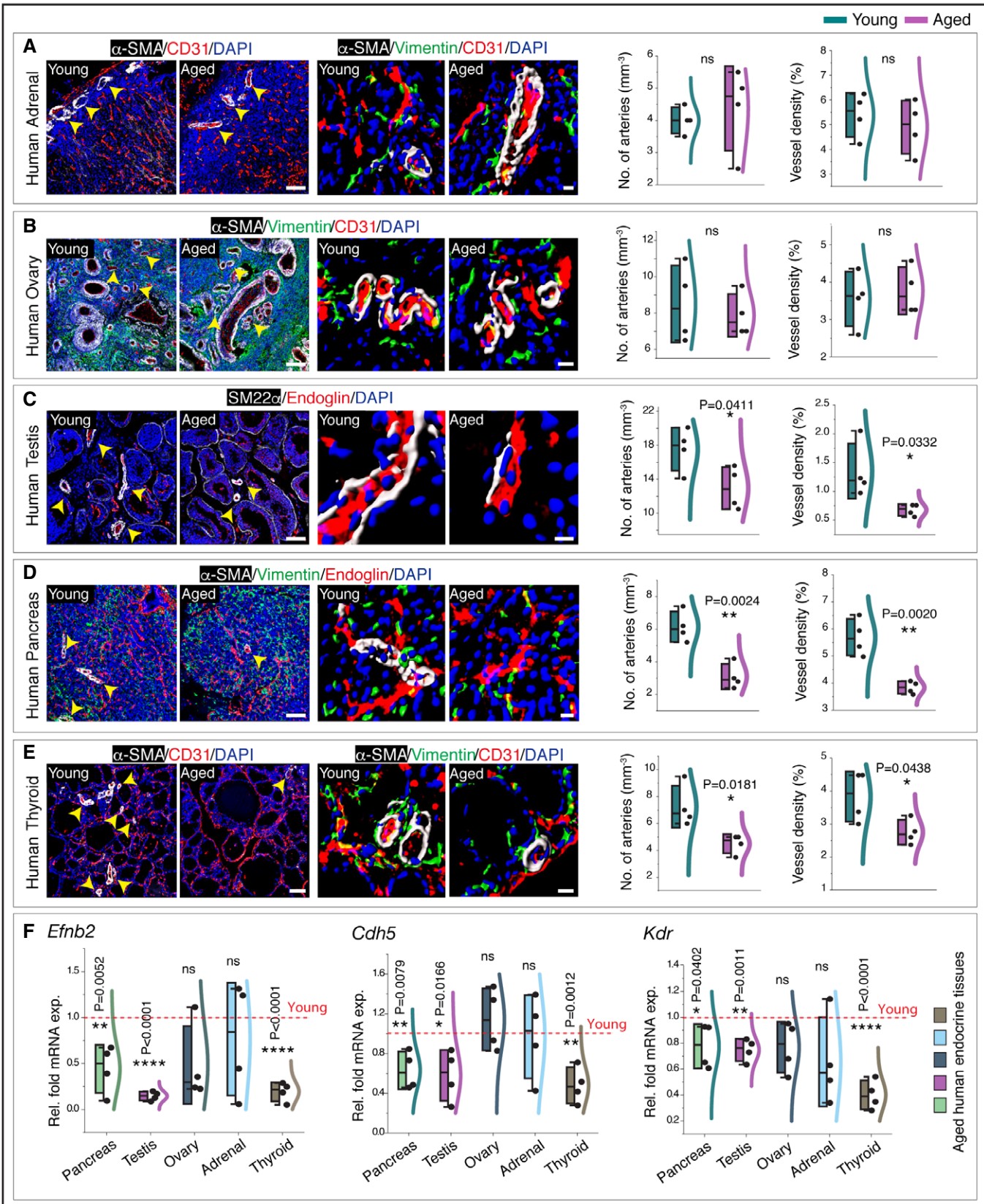

**Figure 4.**

**Figure 4. Vascular and perivascular alterations in the ageing of human endocrine glands.**

A   3D images (left) with α-SMA and CD31 immunostaining in young and aged human adrenal gland. Middle panel images surface rendered for α-SMA, Vimentin and CD31. Arrowheads indicate arteries. Combo plots with quantifications of the number of arteries and vessel density.

B   3D images of young and aged human ovary immunolabelled with α-SMA, Vimentin and CD31. Images in the middle panel with a surface rendered for all the channels. Arrowheads indicate arteries. Combo plots with quantifications of the number of arteries and vessel density.

C   Representative 3D images (left) of young and aged human testis with SM22α and Endoglin immunostaining. Middle panel images surface rendered for all the channels. Arrowheads indicate arteries. Combo plots show quantifications of the number of arteries and vessel density.

D   3D images from young and aged human pancreas staining with α-SMA, Vimentin and Endoglin. 3D images (middle) with a surface rendering for all the channels. Arrowheads indicate arteries. Combined box and whiskers, and scatter plots with the quantifications of the number of arteries and vessel density.

E   Representative 3D images (left) show α-SMA and CD31 immunostaining in young and aged human thyroid gland. Middle panel images surface rendered for α-SMA, Vimentin and CD31. Arrowheads indicate arteries. Combo plots with quantifications of the number of arteries and vessel density.

F   qPCR analysis of *Efnb2*, *Cdh5* and *Kdr* expression (normalized to *Actb*) in aged human endocrine glands compared to young glands.

Data information: (*n* = 4), *P*-values and two-tailed unpaired *t*-tests for all the above panels. ns: not significant; \**P* < 0.05; \*\**P* < 0.01; \*\*\*\**P* < 0.0001. Human samples from donors at ages below 20 years and over 70 years were chosen for young and aged group sets, respectively. Nuclei: DAPI. For the combo plots, the boxes represent mean ± s.d., line in the box is the median, and the lower and upper lines show the minimum and the maximum of the values. The line on the right side of the combo plots represents the sample distribution. Scale bars are 80 μm for 3D images and 10 μm for higher magnification single-cell resolution 3D images.

### Endothelial Gja1 regulates functions of endocrine glands

Vascular changes in $Gja1^{i\Delta EC}$ mutants were supported by reduced numbers of macrophages in the pancreas (Fig EV3F). We analysed other endocrine glands to understand changes in blood vessels and their local microenvironments. Thyroid glands from tamoxifen-treated $Gja1^{i\Delta EC}$ mutant mice showed a significant increase in the blood vessel density (Fig EV4A), together with increases in proliferating ECs as well as a reduction in macrophages (Fig EV4B–D). Additionally, thyroxine hormone, also known as T4 produced by thyroid glands from $Gja1^{i\Delta EC}$ mutant mice, was significantly higher compared to their littermate control mice (Fig EV4A). Similarly, testis from $Gja1^{i\Delta EC}$ mutant mice demonstrated increases in blood vessels and testosterone production (Fig EV4E). Unlike pancreas, testis and thyroid gland, adrenal gland and ovaries from $Gja1^{i\Delta EC}$ mice showed no significant changes (Fig EV5A and B). Therefore, negative regulation on angiogenesis of endothelial Gja1 is restricted to the pancreas, thyroid gland and testis of the endocrine system. Increase in angiogenesis and blood vessels in endothelial-specific Gja1 loss-of-function mice boosts hormone-producing cells—the β cells in pancreases and also hormone production in testes and thyroid glands. In addition to pancreatic β-cell loss with ageing, both T4 and testosterone are known to decrease with ageing (Da Costa *et al*, 2001; Miner & Seftel, 2007; Wylie & Kenney, 2010).

### Reactivation of CD31⁺ Emcnʰⁱ ESM1⁺ ECs in aged mice boosts β-cell proliferation

To further investigate the function of Gja1 on β cells in aged pancreas, aged $Gja1^{i\Delta EC}$ mutant mice were analysed. Following tamoxifen administration and analysis of aged $Gja1^{i\Delta EC}$ mutant mice demonstrated a remarkable increase in CD31⁺ Emcnʰⁱ capillary density within the pancreatic islets compared to the aged littermate control mice (Fig 7K). The reactivation of CD31⁺ Emcnʰⁱ blood vessels within islets was accompanied by an increase in total β-cell numbers in the aged $Gja1^{i\Delta EC}$ mutant mice (Fig 7K). These results of $Gja1^{i\Delta EC}$ mice argue for crucial roles of endothelial Gja1 in controlling pancreatic angiogenesis, islet vessel abundance and β-cell expansion.

Further, for molecular analysis since carbenoxolone (CBX) is a known gap junction inhibitor of Gja1, which induces conformational changes in this protein impairing its function (Schajnovitz *et al*, 2011), we also tested whether CBX promotes the expansion of CD31⁺ Emcnʰⁱ ECs and β-cell expansion in aged animals. While islets of aged, 55-week-old mice treated with vehicle control comprised very few CD31⁺ Emcnʰⁱ vessels, CBX administration led to expansion of CD31⁺ Emcnʰⁱ ECs along with the emergence of ESM-1 expressing cells and increase in β-cell numbers (Fig EV5C–H). Although such drug administration is not endothelial cell specific and may impact other cell types, however, the endothelial phenotype mimics the effects seen in endothelial Gja1 loss-of-function mice. To determine the cause for the loss of proliferation in aged ECs, which showed up-regulation of Gja1, we analysed the expression of cell cycle inhibitors in pancreatic ECs from control and CBX-treated aged mice. This analysis demonstrated a significant up-regulation of p27 by the majority of ECs in aged pancreas (Fig EV5I). However, ECs from CBX-treated mice down-regulated p27 (Fig EV5I). This was accompanied by a significant increase in Ki67⁺ proliferating ECs in aged CBX-treated mice (Fig EV5G and H).

## Discussion

Despite the plethora of data on the physiology and anatomy of the endocrine glands, the cell biology and the tissue microenvironments of the endocrine system have not been studied extensively. Historically, the tissue microenvironments were analysed by electron microscopy and single or dual colour immunohistochemistry on thin paraffin sections (Snippert *et al*, 2011). These methods fail to provide single-cell resolution 3D information and suffer from poor preservation of the tissue architecture. Here, we used an unbiased 3D tissue scanning approach after multiplex immunostaining, which provided new insights into the endocrine tissue microenvironments. All of the information in these datasets cannot be presented in the few figure panels of a manuscript. Moreover, the analysis described above is restricted to our perspective of vascular biology. Therefore, all the raw images are freely available, for viewing and download. This wealth of information will break new grounds for investigations in the various fields of endocrinology, ageing, vascular and matrix biology.

The analysis of major tissue changes in the ageing of the endocrine system reveals that the endocrine tissue and endothelium form a functional unit regulating hormone production. Our endothelial-specific genetic targeting experiments establish that manipulation of

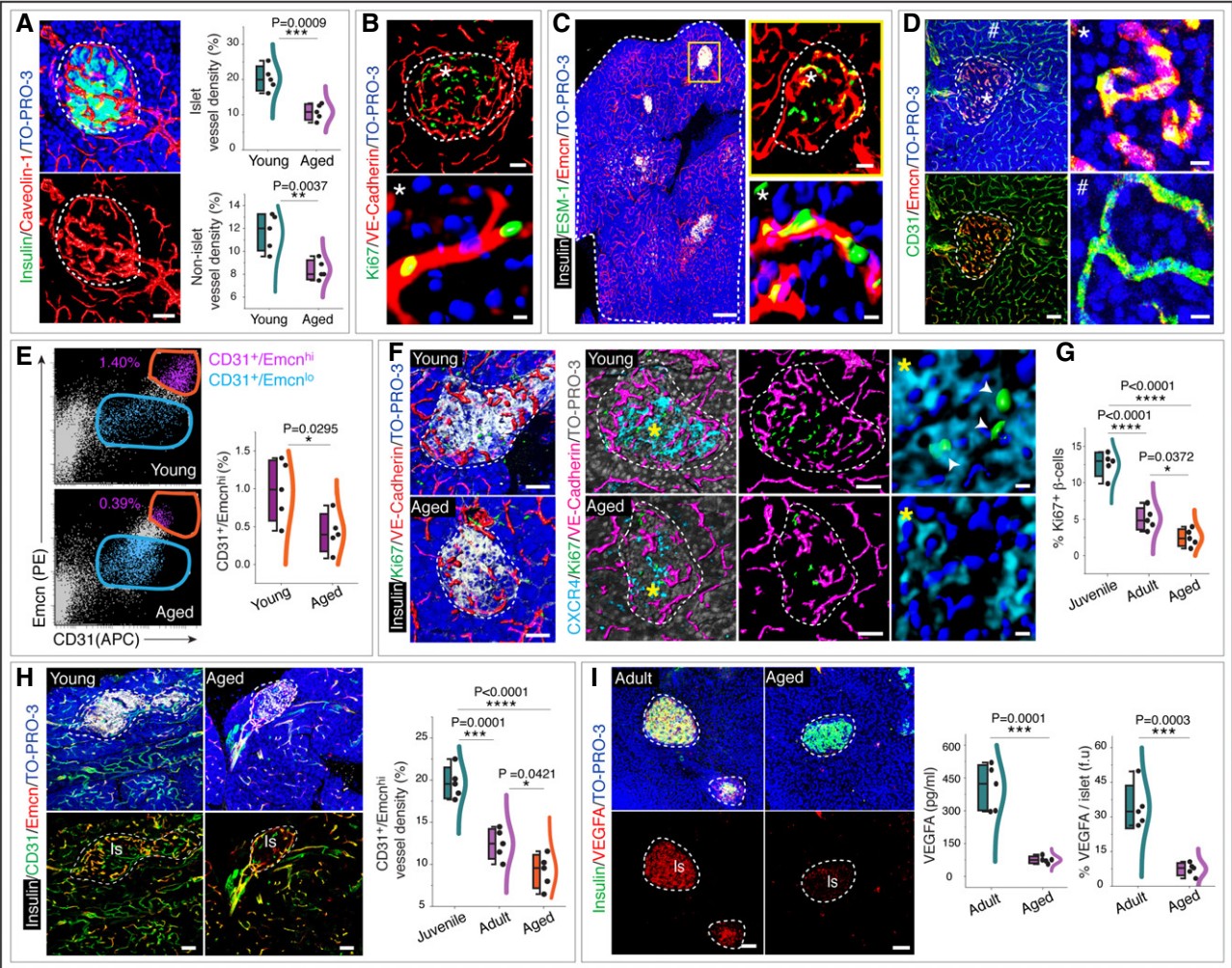

**Figure 5.  Properties of islet endothelium and age-associated modulations.**

A   Representative 3D images with insulin and Caveolin-1 immunostaining in pancreatic islets from young mice. Combined box and whiskers, and scatter plots shows the vessel density analysis in the pancreatic islets and non-islet areas from young and aged mice.

B   Representative 3D images with adult pancreatic islets immunolabelled with Ki67 and VE-Cadherin. Inset (*) shows single-cell resolution and surface rendering for all the channels.

C   Representative tile scan 3D image with insulin, ESM-1 and Emcn immunostaining in 19-week-old mouse pancreas. Higher magnification at the top of the right panel shows the ESM-1+ pancreatic islets. Inset (*) shows single-cell resolution and surface rendering for ESM-1 and Emcn. The white and dashed line in the tile scan image represents the outline of one lobe of the pancreas, and the dashed lines in the inset represent the pancreatic islet.

D   Representative 3D images show CD31 and Emcn expressions in young pancreas. Insets (* and #) show higher magnifications of capillaries in the pancreatic islets and non-islet areas.

E   Representative flow cytometry dot plots showing CD31+ Emcnhi and CD31+ Emcnlo EC populations in young and aged pancreas. Combo plot shows the population of CD31+ Emcnhi ECs in young and aged pancreas.

F   Representative 3D images (left) with insulin, Ki67 and VE-Cadherin immunostaining on young and aged pancreas. 3D images (middle) show CXCR4, Ki67 and VE-Cadherin in young and aged pancreatic islet. Insets (yellow asterisks) show single-cell resolution and surface rendering for Ki67 and CXCR4. Arrowheads represent Ki67+ β-cell.

G   Combo plot shows the quantification of Ki67+ β cells normalized to the total number of β cells per islet in juvenile (3 weeks old), adult (12 weeks old) and aged (55 weeks old) pancreatic islets.

H   Representative 3D images with insulin, CD31 and Emcn immunostaining in young and aged pancreas. Combo plot shows the CD31+ Emcnhi vessel density in juvenile, adult and aged pancreatic islets.

I   Representative 3D images of adult (10 weeks old) and aged pancreas (55 weeks old) with insulin and VEGFA immunostaining. Combo plot (middle) shows the VEGFA as determined by ELISA in isolated islets from 10-week-old adult and 55-week-old aged mice. Combo plot (right) shows the percentage of VEGFA per islet (f.u).

Data information: The white and dashed lines in panels (A, B, D, F, H and I) represent the outlines of pancreatic islets. (n = 5), P-values and two-tailed unpaired t-tests in panels (A, E and I) and one-way ANOVA tests with Tukey's multiple comparisons tests in panels (G and H). *P < 0.05; **P < 0.01; ***P < 0.001; ****P < 0.0001. Nuclei: TO-PRO-3. Is: pancreatic islet. f.u: fluorescence unit. For the combo plots, the boxes represent mean ± s.d., line in the box is the median, and the lower and upper lines show the minimum and the maximum of the values. The line on the right side of these combo plots represents the sample distribution. Scale bars are 200 μm for tile scan 3D images in panel (C), 50 μm for 3D images and 5 μm for the higher magnification single-cell resolution images in panel (B–D and F).

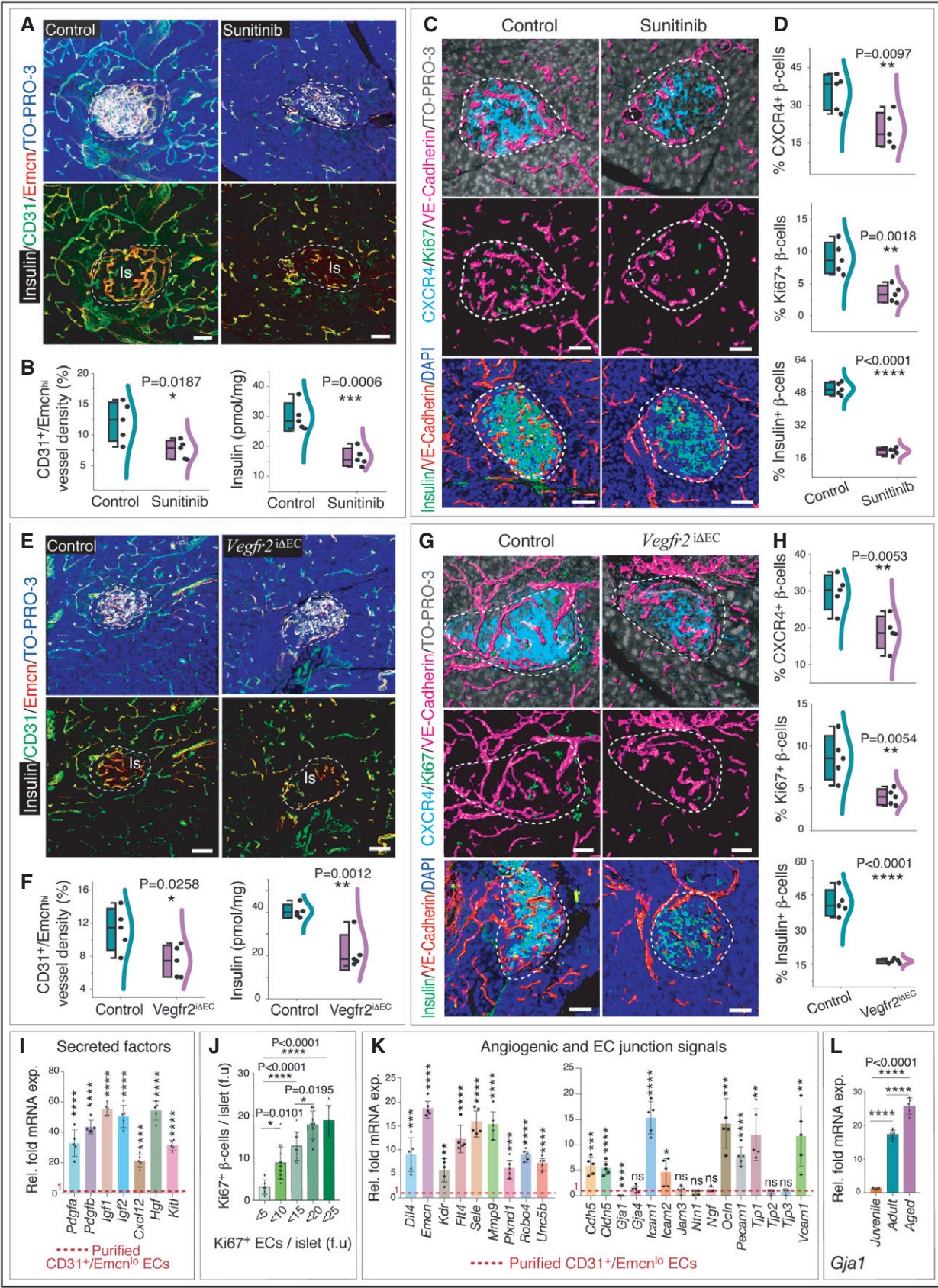

**Figure 6.**

◄

**Figure 6.   Coupling of angiogenesis and β-cell expansion.**

A   Representative 3D images of pancreas from adult (18 weeks old) sunitinib-treated and control mice with insulin, CD31 and Emcn immunostaining.

B   Combined box and whiskers, and scatter plot (left) shows the quantification of CD31$^+$ Emcn$^{hi}$ vessel density in sunitinib-treated and control pancreatic islets. Combo plot (right) shows ELISA-based analysis of insulin.

C   Representative 3D images (upper panel) show CXCR4, Ki67 and VE-Cadherin in sunitinib-treated and control pancreatic islet. 3D images (bottom) show sunitinib-treated and control pancreatic islets immunolabelled with insulin and VE-Cadherin.

D   Combo plot (top) shows the percentage of CXCR4$^+$ β cells per islet in sunitinib-treated and control pancreatic islet. Combo plot (middle) shows the quantification of Ki67$^+$ β cells normalized to the total number of β-cell numbers. Combo plot (bottom) shows the percentage of insulin$^+$ β cells per islet.

E   Representative 3D images of pancreas from adult *Vegfr2*$^{iΔEC}$ mutant and control mice with insulin, CD31 and Emcn immunostainings.

F   Combo plot (left) with quantification of CD31$^+$ Emcn$^{hi}$ vessel density in *Vegfr2*$^{iΔEC}$ and control pancreatic islet. Right combo plot with insulin estimation in pancreas by ELISA.

G   Representative 3D images in the upper panel show CXCR4, Ki67 and VE-Cadherin in islets from *Vegfr2*$^{iΔEC}$ mutant and control mice. 3D images (lower panel) show *Vegfr2*$^{iΔEC}$ mutant and control islets immunolabelled with insulin and VE-Cadherin.

H   Combo plot (top) shows the percentage of CXCR4$^+$ β cells per islet in *Vegfr2*$^{iΔEC}$ mutant and control pancreatic islet. Combo plot (middle) shows the quantification of Ki67$^+$ β cells normalized to the number of β cells. Combo plot (bottom) shows the percentage of insulin$^+$ β cells per islet.

I   Graph shows qPCR analysis of *Pdgfa, Pdgfb, Igf1, Igf2, Cxcl12, Hgf* and *Kitl* expression (normalized to *Actb*) in purified CD31$^+$ Emcn$^{hi}$ ECs compared to CD31$^+$ Emcn$^{lo}$ ECs.

J   Bar graph with the quantification of Ki67$^+$ β-cell numbers per islet (f.u) and Ki67$^+$ EC numbers per islet (f.u). Forty-five pancreatic islets were analysed and quantified from 9 young mice.

K   Bar graphs show qPCR analysis of *Dll4, Emcn, Kdr, Flt4, Sele, Mmp9, Plxnd1, Robo4, Unc5b, Cdh5, Cldn5, Gja1, Gja4, Icam1, Icam2, Jam3, Ntn1, Ngf, Ocln, Pecam1, Tjp1, Tjp2, Tjp3* and *Vcam1* expression (normalized to *Actb*) in purified CD31$^+$ Emcn$^{hi}$ ECs compared to CD31$^+$ Emcn$^{lo}$ ECs from murine pancreas.

L   Graph shows qPCR analysis of *Gja1* expression (normalized to *Actb*) in purified CD31$^+$ Emcn$^{hi}$ ECs from juvenile, adult and aged murine pancreas.

Data information: The white and dashed lines in panels (A, C, E and G) represent the outlines of pancreatic islets. Data represent mean ± s.d. (*n* = 5), *P*-values two-tailed unpaired *t*-tests in panels (B, D, F, H, I and K) and one-way ANOVA tests with Tukey's multiple comparisons tests in panels (J, L). ns: not significant; *\*P* < 0.05; *\*\*P* < 0.01; *\*\*\*P* < 0.001; *\*\*\*\*P* < 0.0001. Nuclei: DAPI or TO-PRO-3 as indicated. Is: pancreatic islet. f.u: fluorescence unit. For the combo plots, the boxes represent mean ± s.d., line in the box is the median, and the lower and upper lines show the minimum and the maximum of the values. The line on the right side of these combo plots represents the sample distribution. Scale bars are 50 μm for all images.

endothelium possesses the potential to correct perturbations in multiple endocrine tissues. Blood vessels have been shown to support β cells and insulin expression in pancreas (Eberhard *et al*, 2010; Almaça *et al*, 2014). However, the tissue-specific features of pancreatic vessels and their role in supporting β-cell expansion and insulin production had remained unknown. Some of these fundamental questions are now resolved in this manuscript. The finding that capillaries in the pancreas are heterogeneous and can be subdivided based on molecular and functional criteria should be beneficial for future investigations.

Angiogenic CD31$^+$ Emcn$^{hi}$ ESM-1$^+$ ECs within islets provide niche signals for β-cell expansion. Further, CD31$^+$ Emcn$^{hi}$ ESM-1$^+$ ECs via their secreted factors and VEGFA expression which is restricted to islets and lost upon ageing are likely to impact the surrounding blood vessels in the exocrine part of the pancreas, ultimately leading to global age-dependent blood vessel decline in pancreas. Our results also

**Figure 7.   Endothelial Gja1 regulates β-cell expansion.**                                                                    ►

A   Representative tile scan 3D images with α-SMA, Isolectin, FABP4 and Emcn in the pancreas from 17-week-old *Gja1*$^{iΔEC}$ mutant and control mice. Insets show higher magnifications of regions.

B   Box whiskers, and scatter plot (left) shows the quantification of CD31$^+$ Emcn$^{hi}$ vessel density in *Gja1*$^{iΔEC}$ mutant and control pancreatic islets. (*n* = 7), *P*-value and two-tailed unpaired *t*-test. Combo plot (right) shows the quantification of P-Histone H3$^+$ EC numbers per islet (f.u) counted on thick murine pancreatic sections. (*n* = 5), *P*-value and two-tailed unpaired *t*-test.

C   3D images show CD31$^+$ Emcn immunostaining in the *Gja1*$^{iΔEC}$ mutant and control pancreas. The scale bars are 50 μm.

D   Combined violin, scatter and sample distribution plot shows the quantification of β-cell mass (%) in pancreatic islets from *Gja1*$^{iΔEC}$ and control mice. Data represent mean ± s.d. (*n* = 140 islets from seven pancreas), *P*-value and two-tailed unpaired *t*-test.

E   Representative 3D images with immunostaining of α-SMA, insulin and VE-Cadherin in the pancreatic islets from *Gja1*$^{iΔEC}$ and control mice. The scale bars are 50 μm.

F   Combo plot (left) shows insulin measurement in lysate of pancreas by ELISA. Combo plot (right) shows the percentage of insulin$^+$ β cells per islet in *Gja1*$^{iΔEC}$ mutant and control mice. (*n* = 5), *P*-values and two-tailed unpaired *t*-tests.

G   Representative 3D images of ESM-1 and Caveolin-1 in pancreatic islets from *Gja1*$^{iΔEC}$ and control mice. Insets (* and #) show higher magnifications with surface rendering for ESM-1 and Caveolin-1 immunostaining. Combo plot with the quantification of ESM-1$^+$ cell numbers per islet (f.u). (*n* = 5), *P*-value and two-tailed unpaired *t*-test. The scale bars for the 3D images are 50 μm, and 5 μm for the higher magnifications with single-cell resolution images.

H   3D images with CXCR4 and Emcn immunostaining in *Gja1*$^{iΔEC}$ and control pancreas. The scale bars are 50 μm.

I   Representative 3D images show CXCR4, Ki67 and VE-Cadherin immunostaining in *Gja1*$^{iΔEC}$ and control pancreas. The scale bars are 50 μm.

J   Combo plots show the percentage of CXCR4$^+$ β cells per islet and Ki67$^+$ β cells normalized to the total number of β cells per islet in *Gja1*$^{iΔEC}$ mutant versus the littermate control mice. (*n* = 5), *P*-values and two-tailed unpaired *t*-tests.

K   Box whiskers, and scatter plot (top) shows the quantification of CD31$^+$ Emcn$^{hi}$ vessel density in aged *Gja1*$^{iΔEC}$ mutant and control pancreatic islets. Data represent mean ± s.d. (*n* = 4), *P*-value and two-tailed unpaired *t*-test. Combo plot (below) shows the quantification of β cells on thick murine pancreatic sections. Data represent mean ± s.d. (*n* = 4), *P*-value and two-tailed unpaired *t*-test.

Data information: The dashed lines in panel A represent the outlines of pancreas. The dashed lines in panels (C, E, G, H and I) represent the outlines of pancreatic islets. Two-tailed Student's *t*-test was performed for the statistical analysis. *\*\*P* < 0.01; *\*\*\*\*P* < 0.0001. Nuclei: DAPI or TO-PRO-3 as indicated. Is: pancreatic islet. f.u: fluorescence unit. The line on the right side of the combo plots represents the sample distribution. Scale bars are 200 μm for tile scan images of longitudinal thick sections across the entire glands, and 50 μm for the higher magnification insets images.

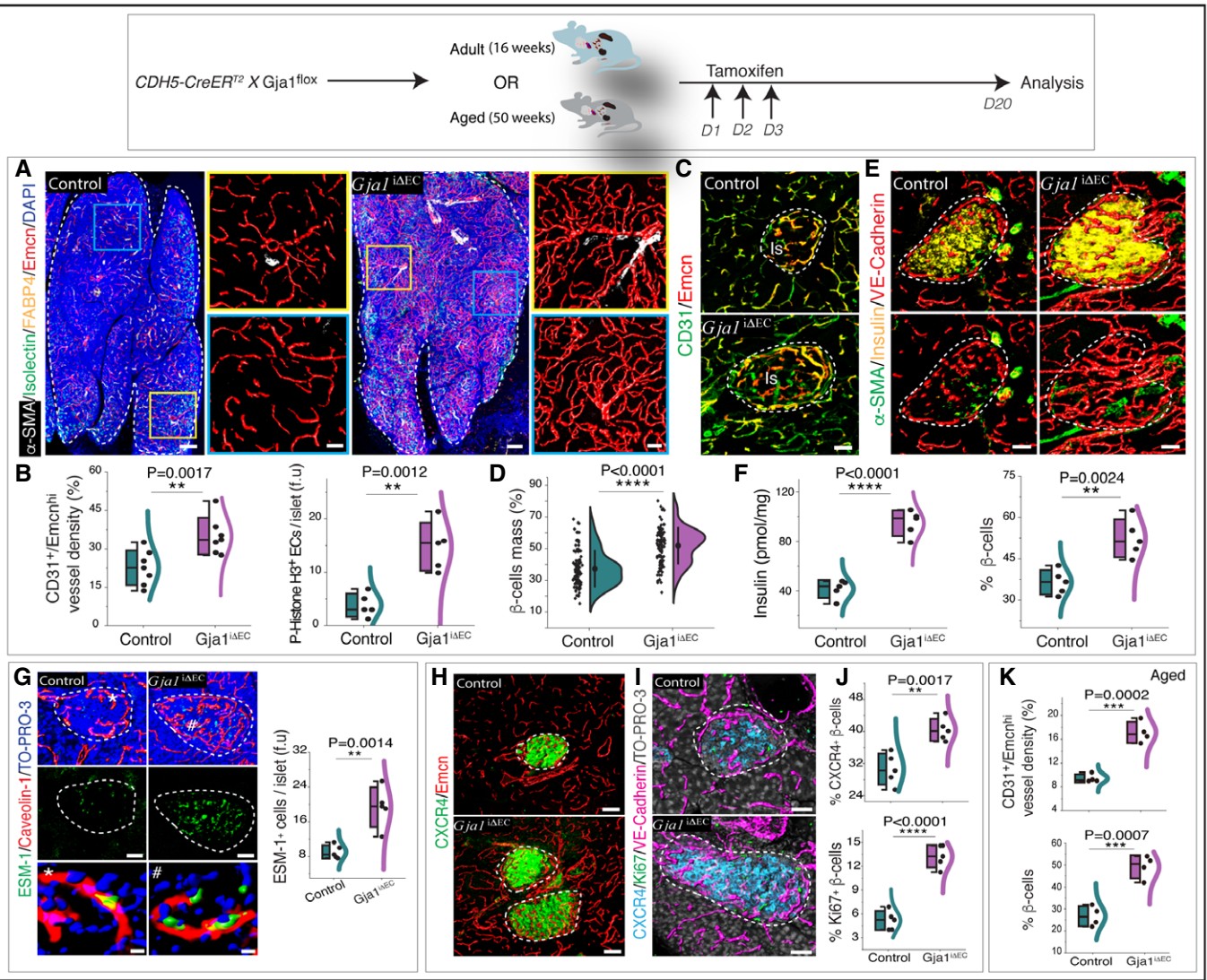

**Figure 7.**

indicate that specific signalling pathways such as gap junction communication can be used to boost angiogenic CD31$^+$ Emcn$^{hi}$ ESM-1$^+$ ECs and thereby β-cell expansion. This could be of great importance in clinical settings, such as the management of diabetes. CD31$^+$ Emcn$^{hi}$ ESM-1$^+$ vessel formation and angiocrine factors for the β cells decline in an age-dependent manner. Specifically, up-regulation of endothelial Gja1 with age led to the decline of these capillaries and active angiogenesis in islets. Previous reports on Gja1 showed that the heterologous deletion of Gja1 diminished angiogenesis in cornea (Rodrigues *et al*, 2010). Also, its expression has been shown in perivascular and stromal cells where it was required for angiogenesis during decidualization (Winterhager *et al*, 2013). Here, we uncover a completely new role of endothelial Gja1 in the ageing of the ECs and its impact on neighbouring cell types. The dramatic positive effect observed after endothelial Gja1 deletion on β-cell mass and expansion within a short time period is exciting from the perspective of achieving β-cell expansion in aged and diabetic individuals. Further, it acts as a negative regulator of angiogenesis in a tissue-specific manner affecting the pancreas, thyroid gland and testis where an age-related decline in vessel density is

observed but not in the ovary or pituitary gland where vasculature does not decline with ageing. Ageing is known to be associated with loss of β-cell self-renewal and numbers (Rankin & Kushner, 2009; Tschen *et al*, 2009; Kushner, 2013). Accordingly, the decline of CD31$^+$ Emcn$^{hi}$ vessels and the concomitant reduction of β-cell expansion could potentially offer a compelling explanation for the loss of β cells in ageing (Wong *et al*, 2009; Chen *et al*, 2011; Kushner, 2013) and might enable therapeutic interventions in elderly and diabetic patients.

# Materials and Methods

### Mice

C57BL/6J mice were obtained from Jackson Laboratory. For wild-type endocrine gland analysis, C57BL/6J mice were used unless stated otherwise. Mice at ages 2–8 and 56–70 weeks were chosen for young and aged group sets, respectively. All the cellular changes were also confirmed by analysing the 10- to 14-week adult mice and comparing them with the aged mice. Before beginning the

experiments, mice were randomly divided into cages and then randomly allocated for treatment. Inducible EC-specific *Gja1* knockout mice (*Gja1*^iΔEC^) were used in this study. In brief, loxP-flanked *Gja1* alleles *Gja1*^lox/lox^ mice were bred with *Cdh5(PAC)-CreERT2* transgenic mice (Wang *et al*, 2010) and then treated with tamoxifen to induce EC-specific *Gja1* gene loss of function. Cre-negative *Gja1*^lox/lox^ mice (control) and Cre-positive *Gja1*^lox/lox^ (*Gja1*^iΔEC^) mutant mice were used in this study. Additionally, transgenic *Vegfr2*^lox/lox^ mice were bred with *Cdh5(PAC)-CreERT2* transgenic mice. Cre-negative *Vegfr2*^lox/lox^ mice (control) and Cre-positive *Vegfr2*^lox/lox^ (*Vegfr2*^iΔEC^) mutant mice were used in this study. Mutant mice were obtained from Jackson Laboratory (008039 for *Gja1*^tm1Dlg^; 018976 for *Kdr*^tm1(cre)Sato^). All animals were genotyped by PCR.

Transgenic and control littermate mice were orally treated with tamoxifen (Sigma, T5648) at a dose of 50 mg/kg body weight for three consecutive days. Vascular network and hormone levels were analysed 14 days after the last dose of tamoxifen.

For metabolic labelling with the hypoxia probe pimonidazole (Hypoxyprobe, Inc., hp15-100kit), young and aged C57BL/6J mice were intraperitoneally injected with 60 mg/kg of pimonidazole. One hour after the injections, mice were sacrificed, and the whole animal perfusion fixation was applied through ascending aorta with 2% paraformaldehyde solution. Then, endocrine glands were collected from young and aged mice. Metabolized Pimo was detected by a rabbit antiserum against the non-oxidized, protein-conjugated form of pimonidazole (Hypoxyprobe).

For sunitinib (Bio Vison, 1611-25) treatment, sunitinib was prepared freshly in sterile water, and adult 16-week-old C57BL/6J mice were injected intraperitoneally at a dose of 60 mg/kg every alternate day for 2 weeks. The same amount of sterile water was administered intraperitoneally in control mice.

For Gja1 inhibitor treatment, CBX (Sigma, C4790) was administered intraperitoneally at a 5 mg/kg dose one time a day for consecutive 10 days in aged C57BL/6J mice. The same volume of sterile water was intraperitoneally injected in age-matched control mice.

All the experiments were performed in accordance with the Home Office Guidance on the Operation of the Animals (Scientific Procedures) Act 1986, published by Her Majesty's Stationery Office (London, United Kingdom). Animals were maintained humanely in compliance with the "Principles of Laboratory Animal Care" formulated by the National Society for Medical Research and the Guide for the Care and Use of Laboratory Animals (National Academies Press, 2011). All animal protocols were approved both by the local University of Oxford or Imperial College London Animal Welfare and Ethical Review Board and by the UK Government Home Office (Animals Scientific Procedures Group).

## Human samples

Formalin-fixed paraffin-embedded (FFPE) standard tissue blocks (pancreas, testis, ovary, thyroid gland and adrenal gland) were purchased from AMS Biotechnology (Oxford, UK). Human specimens provided by AMS Biotechnology (Europe) Limited are legally procured under the laws and regulations of the country. All samples are collected by AMS Biotechnology from consented patients in collaboration with major research/clinical centres under the local EC/IRB-approved protocols. Samples from donors at ages 18–20 and 70–80 years were chosen for young and aged group sets, respectively. Histological study was performed to confirm that sample tissues were healthy and lack disease components. Further details on the healthy human tissue samples are provided (Table EV4). FFPE blocks were kept at −20°C for 30 min before sectioning. Sections with a thickness of 12 μm were generated by microtome blades (FEATHER, 207500006) on a Leica RM2235 microtome.

## Sample preparation for immunostaining

Freshly dissected endocrine glands collected from mice were immediately fixed in ice-cold 2% paraformaldehyde solution for 4 h. Next, the fixed gland tissues were cryoprotected by overnight incubation with 20% sucrose (Sigma-Aldrich, S9378) and 2% polyvinylpyrrolidone (PVP, Sigma-Aldrich, PVP360) solution overnight at 4°C. Then, the tissues were embedded and frozen in 8% gelatin (Sigma-Aldrich, G2625) solution supplemented with 20% sucrose and 2% PVP. Thick sections (100–150 μm) were generated by low-profile blades (Leica, 14035838382) on a Leica CM3050 cryostat. All the endocrine glands were sectioned longitudinally, and sections from the middle/centre of the glands were used, while the sections at the periphery were excluded. Sections were then air-dried for 5 h, frozen and stored until used for immunostainings.

## Immunostaining

For immunostaining, sections were thawed and air-dried for 15 min and hydrated with PBS, followed by permeabilization using 0.3% Triton X-100 for 10 min which was followed by blocking with blocking buffer (5% donkey serum in PBS) at room temperature (RT). Next, the sections were incubated for 4 h at RT or overnight at 4°C with primary antibodies which were diluted in blocking buffer (1:150). After primary antibody incubation, sections were thoroughly washed and incubated with Alexa Fluor-conjugated secondary antibodies (1:300) diluted in blocking buffer for 90 min at RT. Nuclei were counterstained with TO-PRO-3 or 4′,6-diamidino-2-phenylindole (DAPI) as indicated in the figure panels and figure legends. Finally, sections were washed thoroughly with PBS and mounted with Fluoromount-G (Invitrogen, 00-4958-02). After staining, sections were stored at 4°C until imaging was performed. All the slides were imaged within 10 days of staining. No primary antibody during immunostainings were used as negative controls for which tissues were incubated with the antibody diluent alone and no primary antibody, followed by incubation with secondary antibodies.

All the primary antibodies are listed in Appendix Table S1. All the secondary antibodies were used as follows: Donkey anti-Rat IgG Alexa Fluor 594 (A21209, Thermo Fisher Scientific), Donkey anti-Goat IgG Alexa Fluor 488 (A11055, Thermo Fisher Scientific), Donkey anti-Goat IgG Alexa Fluor 647 (A21447, Thermo Fisher Scientific), Donkey anti-Goat IgG Alexa Fluor 546 (A-11056, Thermo Fisher Scientific), Alexa Fluor 488 streptavidin conjugate (S11223, Thermo Fisher Scientific), Alexa Fluor 546 streptavidin conjugate (S11225, Thermo Fisher Scientific), Donkey anti-Rabbit IgG Alexa Fluor 488 (A21206, Thermo Fisher Scientific) and Donkey anti-Rabbit IgG Alexa Fluor 647 (A31573, Thermo Fisher Scientific). Controls for immunostaining on frozen mouse samples were performed without adding primary antibodies but only with secondary antibody incubation.

                                                                  

## Immunohistochemistry on paraffin-embedded human samples

Sections generated from the FFPE blocks of human endocrine glands were firstly deparaffinized in xylene at RT for 3 min with two times, followed by washing in xylene/ethanol (1:1) solution for 3 min. Deparaffinized sections were then dehydrated using graded ethanol solutions and rinsed in cold tap water. Next, all sections were routinely processed for heat-induced antigen retrieval using a universal heat-induced epitope retrieval (HIER) solution (Abcam, ab208572). Briefly, HIER solution was preheated to 95°C using a water bath, and sections were incubated in preheated HIER solution for 10 min, followed by rinsing in distilled water. Next, samples were air-dried for 15 min and processed for immunostaining protocol as described above. Controls for immunostaining on paraffin-embedded human samples were performed without adding primary antibodies but only with secondary antibody incubation.

## Imaging set-up, acquisition and analysis

3D immunofluorescent images for endocrine glands were captured at single-cell resolution (frame size: 1,024 × 1,024 pixels; pixel size: 0.69 μm) on a laser scanning confocal microscope. The imaging equipment set-up consisted of a Zeiss Laser Scanning Microscope (LSM) 880 equipped with seven laser lines (405, 453, 488, 514, 561, 594 and 633 nm), axio examiner (upright) stand and Colibri7 epifluorescence light source with LED illumination, four Objectives, fast scanning stage with PIEZO XY, 32 channels Gallium arsenide phosphide detector (GaAsP) PMT plus 2 channels standard PMT, acquisition and analysis software including measurement, multi-channel, panorama, manual extended focus, image analysis, time lapse, Z Stack, extended focus, autofocus, and with additional modules: Experiment Designer and Tiles and Position. The voxel size for imaging acquisition was 3.2 μm. The imaging and analysis range of z dimension was around 22.14–53.3 μm at 0.82 μm interval size. 20× Plan Apo/0.8 dry lens, 20× Plan Apo 1.0 DIC VIS-IR D0.17 water dipping lens and 10X Plan Apo 0.45 WD = 2.0 M27 dry lens were used for tiling. Large regions through the sections of endocrine gland sections were imaged using tile scan function with appropriate numbers of tiles according to the specific size of each sample, and images were then stitched with 10% overlap using Zen Black (version 3.1, Zeiss) software. All instrument settings were kept at the same between acquisitions of young and aged samples for each gland or between littermate controls and mutants. Z-stacks of images were processed and reconstructed in three dimensions with Imaris software (version 9.2.1, Bitplane). The Imaris Distance Transformation XTensions function was used to analyse the cell–cell interactome (Baccin *et al*, 2019). Maximum of 18 μm distance between the interacting cells was considered within the cell–cell interactome to qualify the interactions. Imaris, Adobe Photoshop and Adobe Illustrator software were used for image processing and analysis in line with The EMBO Journal's guidance for image processing.

## 3D surface reconstruction and cell–cell interactome

For the 3D surface reconstruction of blood vessels, single channel of endothelial markers such as Emcn, Endoglin, Caveolin-1 or VE-Cadherin was reconstructed with surface segmentation using the Surface module in Imaris. First, the region of interest (ROI) was defined and selected. Then, the smoothness of the resulting area was set up using the Smooth option, and the Background Subtraction option was selected for the threshold setting. For the threshold adjustment, the manual option was used to set a proper value according to the surface threshold preview. Next, the resulting reconstruction images were visually inspected to remove small individual segmented components of high sphericity, which were regarded as noise. These components of high sphericity are not caused due to immunostaining or non-specificity of immunostaining or immunostaining artefact but are the part of analysis procedure to remove any noise. For single-cell resolution and surface rendering analysis, higher magnification regions were selected and cropped using 3D crop tool in Imaris. Multiple channels including nuclei, EC, perivascular cells and matrix were reconstructed with 3D surface, as described above. Imaris Distance Transformation XTensions tool was used to calculate the distances between blood vessels and different cell types in order to derive interactive maps and cell–cell interactions.

## Tissue clearing, whole-mount immunostaining and light-sheet microscope-based image acquisition

The tissue clearing and whole-mount immunostaining technique were used based on the PEGASOS method (Jing *et al*, 2018). Briefly, freshly dissected endocrine glands collected from mice were fixed in ice-cold 4% paraformaldehyde. Fixed gland tissues were then decolorized with 25% Quadrol (Sigma-Aldrich, 122262) for 36 h. After thoroughly washing with PBS solution, samples were immersed in the blocking solution composed of 10% dimethyl sulphoxide (Sigma-Aldrich, 276855), 0.5% IgePal630 (Sigma-Aldrich, 18896) and 1× casein buffer (Vector, SP-5020) in PBS solution for 12 h at RT. Next, glands were incubated for 3 days at 4°C with primary antibodies (Emcn and α-SMA) which were diluted in blocking buffer (1:500). After primary antibody incubation, samples were washed with PBS solution for 1 day and stained with secondary antibodies which were diluted in blocking buffer (1:500) for 3 days at 4°C. PBS washing was then performed for 6 h at RT.

Delipidation and dehydration were then performed on samples at RT on a shaker following the passive immersion procedure (30% tert-Butanol [Sigma-Aldrich, 360538] and 3% Quadrol solution for 4 h; 50% tert-Butanol and 3% Quadrol solution for 6 h; 70% tert-Butanol and 3% Quadrol solution for 1 day; 70% tert-Butanol, 30% PEG-MMA-500 [Sigma-Aldrich, 447943] and 3% Quadrol solution for 2 days). Finally, endocrine glands were immersed in BB-PEG clearing medium (75% Benzyl Benzoate [Sigma-Aldrich, B6630], 25% PEG-MMA-500 and 3% Quadrol solution) at RT until transparency was reached. Images of cleared whole-endocrine glands were acquired using PlaneLight, QLS-scope and Zeiss Light Sheet systems, and processed and reconstructed in three dimensions with Imaris software (version 9.2.1, Bitplane).

## Quantifications of imaging data

Blood vessel-related quantification analysis on capillary diameter, artery diameter, vessel density and artery numbers was done with Imaris software (version 9.2.1) or Fiji (version 2.0.0) on single-cell resolution confocal images or light-sheet 3D images. The Imaris

Surface Analysis XTensions tool was used for quantifications of vessel density and artery numbers (Nizari *et al*, 2019). For the analysis of vessel density, both 3D confocal images and whole-organ images acquired by light-sheet microscope were used. The region of the whole tissue was selected using Crop 3D tool, and the total tissue volume was acquired via Volume Statistics function in Imaris. Then, a single channel of endothelial marker was reconstructed in 3D using Surface function, and the tissue volume of blood vessels was measured using Surface Statistics function. Same threshold parameters were applied when comparing young and aged samples or when comparing the samples within each analysis. The vessel density was calculated by dividing the tissue volume of blood vessels in the numerator by the total tissue volume in the denominator. For the analysis and quantifications of the artery numbers, whole-endocrine gland light-sheet images were used, and the total tissue volume was acquired via Volume Statistics function in Imaris. A single channel of α-SMA (Sigma, C6198) was reconstructed, and arteries were then distinguished based on their structure and tubular shape and counting was performed. Same threshold parameters were applied when comparing young and aged samples for each endocrine gland. The artery numbers per 1 mm$^3$ tissue area were calculated by dividing the total number of arteries in the numerator by the total tissue volume in the denominator. For the quantifications of capillary diameters, Emcn (Santa Cruz, sc-65495) or Endoglin (R&D system, AF1320) was used and for artery diameter analysis α-SMA (Sigma, C6198), immunostaining was used. Since α-SMA could also be expressed by some stromal cells, arteries were distinguished based on their tubular shape. Briefly, for capillary and artery diameter quantifications, a single channel of Emcn/Endoglin/α-SMA was reconstructed using Z-project function with max intensity mode. Seven random regions in each section of the gland tissue were selected for quantifications of capillary and artery diameters. Average diameter of capillaries and arteries in each sample was then calculated according to the measurements using the distance tools of Fiji. For the analysis of CD31$^+$ Emcn$^{hi}$ vessel density in pancreas, the pancreatic islet regions in pancreas were selected based on insulin expression, and vessel density was calculated as described above.

Quantification of β cells was performed based on insulin and CXCR4 expressions. Multiple thick sections which spanned the entire pancreas were used for quantifications. Regions of pancreatic islet on thick sections were selected for each pancreas using Fiji, and the insulin- or CXCR4-stained area were measured within the pancreatic islet. The percentage of insulin$^+$ β-cell and CXCR4$^+$ β-cell was calculated by dividing the insulin- or CXCR4-stained area in the numerator by the total area of the selected pancreatic islet in the region defined as referent.

For quantification of β-cell mass in the *Gja1*$^{iΔEC}$ mutant and littermate control mice, numerous thick sections which spanned the entire pancreas were used and 160 islets were counted. Briefly, regions of pancreatic islet were selected for each pancreas using Fiji (version 2.0.0), and the CXCR4- or insulin-stained area were measured within the pancreatic islets to calculate the β-cell mass.

For quantifications of follicle density, after 3D reconstruction using Imaris, the follicle density of thyroid gland and ovary was calculated by dividing the number of follicles in the numerator by the total area of the whole gland in the denominator defined by the expression of nuclear stain.

Nuclei detection and membrane detection function in the Cell module of Imaris were used to automatically segment and analyse the numbers of CD68$^+$ macrophages and FSP1$^+$ fibroblasts. Under the Cell Creation wizard of Imaris, for CD68$^+$ macrophages, CD68 channel was selected as source channel for membrane-based detection and TO-PRO-3 channel was selected for nuclei-based detection; for FSP1$^+$ fibroblasts, FSP1 and TO-PRO-3 channels were selected for membrane-based detection and nuclei-based detection, respectively. Nuclei locations were used as seed points for an algorithm performing a cell membrane calculation, which was used to distinguish between the inner and outer boundaries of the cell. The final results displayed accurate segmentation of cells and presented the total number of CD68$^+$ macrophages and FSP1$^+$ fibroblasts. Next, the number of CD68$^+$ macrophages and FSP1$^+$ fibroblasts per tissue volume (mm$^3$) was calculated by dividing the number of segmented cells in the numerator by the total tissue volume in the denominator.

For quantifications of percentage of VEGFA coverage per islet (f.u) in pancreas, islet was identified and selected based on insulin immunostaining, and the percentage of VEGFA coverage was calculated by dividing the VEGFA-stained area in the numerator by the total area of the selected pancreatic islet in the region defined as referent.

### Cell proliferation analysis and tip cell quantifications

Quantifications of the number of proliferative cells and tip cells were conducted with Imaris or Fiji software. Phospho-Histone H3 (Merck, 631257) or Ki67 (Abcam, ab15580) immunostaining was performed to identify proliferating cells. ESM-1 (R&D systems, AF1999) immunostaining was performed to identify tip cells, and Emcn was used to identify blood vessels/ECs in pancreas. CXCR4 (Abcam, ab1670) immunostaining was applied to identify β cells in pancreatic islet. Nuclei were counterstained with TO-PRO-3.

For quantifications of ESM-1$^+$ cell numbers per islet (f.u), pancreatic islet regions were selected for each pancreas sample to acquire the average measurement, and the number of ESM-1$^+$ cells was counted using Fiji by merging the channels of TO-PRO-3, ESM-1 and Emcn on thick sections. For quantifications of Phospho-Histone H3$^+$/Ki67$^+$ EC numbers per islet (f.u), regions of pancreatic islet were selected for each pancreas, and the number of Phospho-Histone H3$^+$ or Ki67$^+$ ECs was counted by merging the channels of TO-PRO-3, Phospho-Histone H3/Ki67 and Emcn. For these quantifications, several tissue sections were analysed from each pancreas. Nuclei detection and membrane detection function in the Cell module of Imaris were used to automatically segment and analyse the Ki67$^+$ β cells and CXCR4$^+$ β cells. The percentage of Ki67$^+$ β-cell numbers was calculated using the number of segmented Ki67$^+$ β cells after normalization to the total number of segmented β cells in each pancreatic islet.

### Image database

Images are available, on via an OMERO web interface (Allan *et al*, 2012; Burel *et al*, 2015) which enables easy online access to view image sets, basic 2D analysis capabilities and the ability to download image sets for offline advanced analyses such as 3D surface rendering. This interface allows viewing and download of the images which then can be subjected to 3D rendering and further analysis. The link to access image database is http://homeros.kennedy.ox.ac.uk/pub/chen-et-al-2020-3dEndocrine

### Flow cytometry

For flow cytometry, C57BL/6J mice were used for collecting pancreatic cells. Pancreases were crushed in 0.7 mg/ml collagenase A (Sigma, 10103578001) and incubated at 37°C for 45 min. After the digestion, pancreatic cells were added with 100 µl FBS and filtered to obtain a single-cell suspension. The equal number of cells was obtained by counting. Then, cells were stained with Emcn on ice for 2 h. After washing twice with cold PBS, cells were stained with APC-conjugated CD31 (R&D System, FAB3628A) and PE donkey anti-rat secondary antibody (Thermo Fisher, 712-116-153) on ice for 2 h. After washing thoroughly, cells were acquired using Fortessa X20 flow cytometer and data were analysed via BD FACSDiva software (version 6.0, BD Bioscience).

For cell sorting, freshly dissected pancreases were subjected to collagenase digestion to prepare single-cell suspension for the isolation of ECs. Pancreas were crushed in 0.7 mg/ml collagenase A and incubated at 37°C for 45 min. ECs from the single-cell suspension were isolated using a magnetic bead-based separation method (Dynabeads: Sheep Anti-Rat IgG; 11035) using rat CD144 and Emcn antibodies.

For FACS analysis of p27, following the bead-based sorting of ECs from young and aged pancreases, cells were then fixed in 2% ice-cold paraformaldehyde for 20 min at 4°C. Cells were again washed and then permeabilized in 0.4% (v/v in PBS) saponin (Sigma, 47036) for 20 min before being washed again in PBS/0.1% Na azide/0.1% BSA and incubated with specific antibody to p27 (PA5-16717, Thermo Fisher Scientific) or isotype-matched negative control antibodies for 1 h at 4°C. Unbound antibody was removed by washing in PBS/0.1% BSA. Cells were then incubated for 1 h at 4°C with FITC-conjugated secondary antibody before being washed and analysed by flow cytometry.

### Quantitative reverse transcription PCR

For the analysis of mRNA expression levels in EC subsets, $CD31^+$ $Emcn^{hi}$ and $CD31^+$ $Emcn^{lo}$ mouse pancreatic cells were sorted by fluorescence-activated cell sorting (FACS) directly into the lysis buffer of the RNeasy Mini Kit (QIAGEN, 74104). Total RNA was isolated according to the manufacturer's protocol. A total of 100 ng RNA per reaction was used to generate cDNA with the iScript cDNA Synthesis System (Bio-Rad, CT011243). qPCR was performed using TaqMan gene expression assays on ABI PRISM 7900HT Sequence Detection System. The FAM-conjugated TaqMan probes were used along with TaqMan Gene Expression Master Mix (Applied Biosystems, 4369510). Gene expression assays were normalized to endogenous VIC-conjugated *Actb* probes as standard. For analysis of mRNA expression levels from FPPE blocks of human endocrine tissues, 25 µm sections were generated from FPPE blocks of tissues with human endocrine glands. All FFPE sections were deparaffinized involving immersion into xylene twice followed by immersion into 100% ethanol twice then left to air dry for 15 min. RNA extraction was performed using FFPE RNA extraction kit (QIAGEN, 73504) according to the manufacturer's instructions. All RNA samples were immediately processed for cDNA preparation using Super-Script IV First-Strand Synthesis System (Invitrogen, 18091200). FAM-conjugated TaqMan probes were used along with TaqMan Gene Expression Master Mix (Applied Biosystems, 4369510) to perform qPCR.

### Islet isolation from mouse pancreas

The protocol of islet isolation was used based on previous report (Li *et al*, 2009). Briefly, pancreas perfusion was performed by injecting 3 ml Liberase (Roche Diagnostics, 5401127001) solution through the joint site of the hepatic duct and the cystic duct and reach the middle of common bile duct under the microscope. Pancreas was then removed and digested at 37°C for 15 min. After centrifuging at 290 g for 30 s at 4°C, the supernatant was discarded, and resulting pellet was resuspended and then filtered with 70 µm cell strainer with HBSS solution supplemented with $CaCl_2$ (1 mM). The strainer was turned upside down over a new petri dish and rinsed the captured islets into the dish. Then, the isolated islets were hand-picked using a pipette with a wide-open tip.

### ELISA

For tissue lysate preparation, 300 µl complete extraction buffer (100 mM Tris [Sigma, T1503], 150 mM NaCl [Sigma, S7653], 1 mM EGTA [Sigma-Aldrich, E4378], 1 mM EDTA [Sigma, E6758], 1% Triton X-100 [VWR, 306324N] and 0.1% Sodium deoxycholate [Sigma-Aldrich, D6750]) supplemented with protease inhibitor cocktails was added into 5 mg tissue slices. Then, samples were homogenized with an electric homogenizer and maintained at constant agitation for 2 h at 4°C. After centrifuging for 20 min at 19,000 g at 4°C, the samples were placed on ice. The lysate was aliquoted and stored at −80°C. Insulin (Thermo Fisher Scientific, EMINS), thyroxine (Thermo Fisher Scientific, EIAT4C) and testosterone (Abcam, ab108666) in the tissue lysate were determined by enzyme-linked immunosorbent assay (ELISA) Kits based on the manufacturer's instructions. VEGFA was determined on lysate from isolated islets derived from young and aged mice using ELISA kit (Novus Biologicals, NBP1-92679) based on the manufacturer's instructions.

### Statistical analysis

All data are presented as mean ± s.d. For analysis between two groups, the significance of the difference in mean values was determined using two-tailed Student's *t*-test, unless indicated otherwise. For analysis of the statistical significance of differences between more than two groups, we performed one-way analysis of variance (ANOVA) test with Tukey's multiple comparisons test. $P < 0.05$ was considered significant. ns, not significant; *$P < 0.05$; **$P < 0.01$; ****$P < 0.001$; ****$P < 0.0001$. All statistical analyses were performed using GraphPad Prism software (version 8.02). Graphs and combo plots were made using the OriginPro (version 9.1) software. No randomization or blinding was used, and no animals were excluded from analyses. Sample sizes were selected on the basis of previous experiments. Combined box and whiskers, and scatter plot: in each graph, the boxes represent mean ± s.d., line in the box is the median, and the lower and upper lines show the minimum and the maximum of the values. The line on the right side of these combo plots represents the sample distribution. For the combined violin and scatter plots, the line in the violin represents mean ± s.d., and the dot in the midline represents mean. Several independent experiments were performed to guarantee reproducibility of findings.

## Data availability

The image database from this publication has been deposited to the OMERO server https://www.openmicroscopy.org/omero/ and assigned the link: http://homeros.kennedy.ox.ac.uk/pub/chen-et-al-2020-3dEndocrine.

**Expanded View** for this article is available online.

## Acknowledgements

We acknowledge the Kennedy Institute animal facility staff for technical assistance and tamoxifen injections. We thank Dr. Xi Hu (PlaneLight) for supporting the light-sheet microscope image acquisition. We thank Mr. Pengjun Xi for technical assistance. A.P.K is supported by Medical Research Council (CDA: MR/P02209X/1), European Research Council (StG: metaNiche, 805201), Leuka (2017/JGF/001), The Royal Society (RG170326), Kennedy Trust for Rheumatology Research (KENN 15 16 09), CRUK Development Fund (CRUKDF 0317-AK) and John Fell OUP Research Fund (161/061). This work is also supported by the Sir Henry Dale Fellowship (202300/Z/16/Z) from the Wellcome Trust and the Royal Society to S.K.R; and the Wellcome Trust PRF (100262Z/12/Z) and a Kennedy Trust for Rheumatology Research grant to M.L.D.

## Author contributions

JC, SKR and APK designed and organized the experiments, and interpreted results. JC and LL performed most experiments and analysed data. SLT performed the imaging experiments. JC assembled the data. JC and RL prepared figures. BDM provided support for data storage and OMERO data presentation. MLD contributed to the design of imaging set-up, provided advice and commented on the manuscript. JC and APK revised the manuscript. APK and JC wrote the paper. APK conceived, devised and supervised the study.

## Conflict of interest

The authors declare that they have no conflict of interest.

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
