## [Review Process File · The EMBO Journal]

Decreased blood vessel density and endothelial cell subset dynamics during ageing of the endocrine system

Anjali Kusumbe, Junyu Chen, Luciana Lippo, Rossella Labella, Sin Lih Tan, Brian Marsden, Michael Dustin, and Saravana Ramasamy

DOI: [10.15252/embj.2020105242](https://doi.org/10.15252/embj.2020105242)

Corresponding authors: Anjali Kusumbe (anjali.kusumbe@kennedy.ox.ac.uk)

Review Timeline:

Submission Date:	9th Apr 20
Editorial Decision:	13th May 20
Revision Received:	11th Aug 20
Editorial Decision:	1st Sep 20
Revision Received:	24th Sep 20
Accepted:	28th Sep 20

Editor: Daniel Klimmeck

Transaction Report:

Dear Dr Kusumbe,

Thank you for the submission of your manuscript (EMBOJ-2020-105242) to The EMBO Journal. Your manuscript has been sent to three reviewers, and we have received reports from all of them, which I enclose below.

As you will see, the referees acknowledge the potential novelty of your results and value as a resource to the field, although they also express a number of issues that will have to be conclusively addressed before they can be supportive of publication of your manuscript in The EMBO Journal. The reviewers raise a number of points related drugs used (ref#3), data presentation aspects and discussion of the findings as well as preceding literature, which would need to be conclusively addressed to achieve the level of robustness and clarity needed for The EMBO Journal.

I judge the comments of the referees to be generally reasonable and given their overall interest, we are in principle happy to invite you to revise your manuscript experimentally to address the referees' comments.

Please note that while the 3D clearance concern of referee #2, (an imaging expert) is well taken, it is not overriding in our view, however we agree it will be important to better consider this limitation and discuss its consequences.

Further, please note that the online interface for data browsing should be made accessible for referees and readers.

Please let me know any time if you have additional questions or need further input on the referee comments.

Please see below for additional instructions for preparing your revised manuscript.

Thank you for the opportunity to consider your work for publication. I look forward to your revision.

Kind regards,

Daniel Klimmeck

Daniel Klimmeck, PhD

Editor
The EMBO Journal

Before submitting your revision, primary datasets (and computer code, where appropriate) produced in this study need to be deposited in an appropriate public database (see <https://www.embopress.org/page/journal/14602075/authorguide#datadeposition>).

The accession numbers and database should be listed in a formal "Data Availability" section (placed after Materials & Method) that follows the model below (see also <https://www.embopress.org/page/journal/14602075/authorguide#availabilityofpublishedmaterial>). Please note that the Data Availability Section is restricted to new primary data that are part of this study.

Data availability

Our journal also encourages inclusion of *data citations in the reference list* to directly cite datasets that were re-used and obtained from public databases. Data citations in the article text are distinct from normal bibliographical citations and should directly link to the database records from which the data can be accessed. In the main text, data citations are formatted as follows: "Data ref: Smith et al, 2001" or "Data ref: NCBI Sequence Read Archive PRJNA342805, 2017". In the Reference list, data citations must be labeled with "[DATASET]". A data reference must provide the database name, accession number/identifiers and a resolvable link to the landing page from which the data can be accessed at the end of the reference. Further instructions are available at <https://www.embopress.org/page/journal/14602075/authorguide#referencesformat>

- a point-by-point response to the referees' comments, with a detailed description of the changes made (as a word file).

- a word file of the manuscript text.
 - individual production quality figure files (one file per figure)
 - a complete author checklist, which you can download from our author guidelines (<http://emboj.embopress.org/authorguide>).
 - Expanded View files (replacing Supplementary Information)
- Please see out instructions to authors
<https://www.embopress.org/page/journal/14602075/authorguide#expandedview>

The revision must be submitted online within 90 days; please click on the link below to submit the revision online before 11th Aug 2020.

Link Not Available

Referee #1:

This manuscript explores age-related changes which occur in mouse and human endocrine glands. The study combines a comprehensive "systems imaging" approach for analysis of age-related effects on endocrine tissues with a more targeted gene modification study of the role of the gap junction protein Gja1 in age-related vascular changes. The latter study provides a compelling example of the biological insight which can be gained by leveraging information generated by the systems imaging. The data are extensive, novel and informative, and the manuscript is well presented and clearly written. The data provide solid support for the conclusions which are made. Overall, the datasets provided should be a useful resource for researchers in a range of areas including endocrinology and vascular biology. I do not have any suggestions for modification of the manuscript.

Referee #2:

Manuscript Summary:

Chen et al. uses 3D imaging to describe age-dependant vascular changes in the endocrine system. Tissue sections are immunostained and imaged in 3D, upon which they are quantified. Unfortunately, the large amount of different samples reflects the authors focus on quantity over quality. Many of the age related processes observed in this manuscript are not novel, and in addition they are not very well quantified from a 3D microscopy perspective. It therefore does not

provide novel insights nor interesting methods to the field. The data is shared as a resource, which is commendable, especially with the screening of 150 antibodies, valuable for the imaging field. However, the limited depth of the imaging severely reduces the importance of these data as a 3D resource.

Major Concerns:

1. The authors did not use any optical clearing, thereby severely limiting their 3D sampling volume. From the imaging data it often looks like only a few layers of cells are imaged. This is apparent at the end of the thyroid movie 6.
2. Fig2. Vessel density is calculated as a volume percentage. Without optical clearing, the signal intensity will steeply decline in the z dimension, influencing the threshold-based rendering of the vessels (they will not render deeper in the sample, where signal is low). In addition, from the methods it looks like the authors did not use a single threshold for all samples. Possibly resulting in large biases when comparing rendered volumes between samples. This is a concern in all volume calculations performed by the authors.
3. Artery number potentially suffers from the lack of 3D size. Arteries are manually counted and presented normalized to the volume. But an artery might run in and out of the small sampled volume, resulting in an inaccurate estimation of the artery number. An example can clearly be seen in Fig2 panel E.
4. Fig4. From the methods it becomes apparent that Ki67 quantification is done by manually counting ki67+ b-cells and the total number of cxcr4+ b-cells within each pancreatic islet. However, the authors should mention of the total number of counted cells on which this quantification is based. In addition, manually counting large numbers of cells in 3D is error prone and therefore usually performed by segmentation using imaging analysis software. This should be straightforward to do for the authors as they claim to have single cell resolution.
5. The comments in 4 also apply to the manual counting of macrophages.

Minor comment:

The Imaging set-up, acquisition and analysis section of the methods does not mention the resolution and range of the z dimension. The authors should mention the voxel size instead of the pixel size, and mention the imaged/analysed depth for the samples.

Referee #3:

Chen et al., present an impressive imaging encyclopedia of endocrine organs, ovary, pituitary, adrenal glands, pancreas, thyroid, testis, from young and old mice as well as humans. Very interestingly, there seems to be similarities in patterns of how vessel density and number of arteries change from young to aged between mice and humans. The images performed with a wide range of antibodies in the different organs, ages and species have been deposited at Omero, an imaging repository. This is highly commendable and points to future new development in how to work with large scale images. Moreover, the authors have put a lot of effort into the presentation, which allows the reader to follow and draw conclusions even though the amount of data is close to overwhelming. Overall, this is a paper which will inspire to a wide range of exciting, hypothesis-driven research both in the authors' own laboratory and in many others.

Comments

1. As far as this reviewer could find, there was no access to files at Omero and so I have not been able to explore the potential use for the community and in particular, the user-friendliness.
2. Figure 1. Please place the tile scan of each organ with the "representative" 3D image. Or at least show the organs in the same order (in all relevant figures). Alternatively, remove the 3D images (not so useful) and just to show the tile scans, which are more interesting.
3. The schematics in Figure 2 panel M is important but it is very busy as shown. Some cells are shown twice such as CD68. The two CD68 cells appear to have different relations to the medulla and transition zones - one cell has a line drawn for the young tissue but not the other. Can this be simplified/improved? Perhaps young on one side and aged on the other of the schematics. Alternatively, for each zone (medulla, transition, cortex), show all relevant cells and thereby avoid the busy lines. Perhaps the different mural cells on one side and inflammatory cells on the other? Please indicate which vessel type is depicted here. This is an important and very useful summary schematic and should be given more space. It's very interesting to see how the mural cell composition differs between the different sections.
4. Figures 2 and 3; show the organs in the same order as for Fig. 1. For Figure 2, please specify how arteries were identified in the different organs. For Figure 3A, the heatmap of murine organs should not be in this figure? Moreover, considering the large variability within each organ, between vessel types etc, how have the authors come to conclude that expression of a particular marker is high or low on a single cell level rather than in average? This does not make sense as shown.
5. In Figures 4 and 5, the authors define the vasculature in the islets of Langerhans as CD31+, Emcn high while the exocrine vasculature is Emcn low. The Emcn high vasculature diminishes with age and is sensitive to VEGFR2 inhibition with Sunitinib. Although an elegant application of findings from the broad imaging endeavor, these data are not very novel but already shown by a range of different laboratories. In relation to the previous literature, it would be useful to also show expression of PLVAP as a marker of fenestrated vessels, known to be lost upon VEGFA deprivation (see for example Kamba et al. *Am J Physiol Heart Circ Physiol.* 2006 Feb;290(2):H560-76).
6. In Figures 6 and 7, the authors focus on connexin43 (Gaj1) in the pancreas vasculature, as Gaj1 increases with age. Specific deletion of Gaj1 in endothelial cells leads to a range of seemingly favorable changes such as increased vessel density and proliferation of b-cells in Langerhans islets and increased insulin production. Treatment with Carbenoxolone (CBX), a drug said to decrease expression of Gaj1, mimicks the effects of genetic deletion. CBX does not have a good reputation as a drug and is known to act on a range of pathways including NMDA receptors, GABA A receptors, and calcium channels. It would probably be best to take the CBX data out; they are not needed.

Point-by-point response

We are very grateful to the editor and the reviewers for their time and valuable comments. We have substantially revised the manuscript to address all the comments and suggestions. This has helped to improve the manuscript further. All the changes are highlighted in the manuscript.

Referee #1:

This manuscript explores age-related changes which occur in mouse and human endocrine glands. The study combines a comprehensive "systems imaging" approach for analysis of age-related effects on endocrine tissues with a more targeted gene modification study of the role of the gap junction protein Gja1 in age-related vascular changes. The latter study provides a compelling example of the biological insight which can be gained by leveraging information generated by the systems imaging. The data are extensive, novel and informative, and the manuscript is well presented and clearly written. The data provide solid support for the conclusions which are made. Overall, the datasets provided should be a useful resource for researchers in a range of areas including endocrinology and vascular biology. I do not have any suggestions for modification of the manuscript.

We are very grateful to the reviewer for the positive comments and appreciation of the work. Thank you very much for acknowledging the novelty of the study and appreciating data presentation and associated information in the study.

Referee #2:

Manuscript Summary:

Chen et al. uses 3D imaging to describe age-dependant vascular changes in the endocrine system. Tissue sections are immunostained and imaged in 3D, upon which they are quantified. Unfortunately, the large amount of different samples reflects the authors focus on quantity over quality. Many of the age related processes observed in this manuscript are not novel, and in addition they are not very well quantified from a 3D microscopy perspective. It therefore does not provide novel insights nor interesting methods to the field. The data is shared as a resource, which is commendable, especially with the screening of 150 antibodies, valuable for the imaging field. However, the limited depth of the imaging severely reduces the importance of these data as a 3D resource.

Major Concerns:

1. The authors did not use any optical clearing, thereby severely limiting their 3D sampling volume. From the imaging data it often looks like only a few layers of cells are imaged. This is apparent at the end of the thyroid movie 6.

Thank you for the comment. For all the images throughout the study, numerous layers were acquired. For example, for Fig. 1A with the section gallery view (as shown below) 27-40 layers were acquired.

Also, we have now performed light-sheet imaging of the cleared whole glands, particularly for analysing the vessel density and artery numbers (Fig 3A-E, Movies EV7-11) and the related method has been added to the revised manuscript. We observe similar changes in vessel densities and artery numbers across young and aged glands using thick sections or cleared whole glands. (Page 7, 24 and 25)

2. Fig2. Vessel density is calculated as a volume percentage. Without optical clearing, the signal intensity will steeply decline in the z dimension, influencing the threshold-based rendering of the vessels (they will not render deeper in the sample, where signal is low). In addition, from the methods it looks like the authors did not use a single threshold for all samples. Possibly resulting in large biases when comparing rendered volumes between samples. This is a concern in all volume calculations performed by the authors.

We thank the reviewer for raising these points.

Same threshold parameters were used when comparing young and aged glands or other samples within each analysis. We have added this information to the revised manuscript. (Page 25 and 26).

In addition, we have now performed light-sheet imaging of the cleared whole glands, particularly for analysing the vessel density and artery numbers (Fig 3A-E, Movies EV7-11) and the related method has been added to the revised manuscript. We observe similar changes in vessel densities and artery numbers across young and aged glands using thick sections or cleared whole glands. (Page 7, 24 and 25)

3. Artery number potentially suffers from the lack of 3D size. Arteries are manually counted and presented normalised to the volume. But an artery might run in and out of the small sampled volume, resulting in an inaccurate estimation of the artery number. An example can clearly be seen in Fig2 panel E.

Thank you for this comment. As mentioned above, we have now performed light-sheet imaging of the cleared whole glands, particularly for analysing the vessel density and artery numbers (Fig 3A-E, Movies EV7-11) and the related method has been added to the revised manuscript. We observe similar changes in vessel densities and artery numbers across young and aged glands using thick sections or cleared whole glands. (Page 7, 24 and 25)

4. Fig4. From the methods it becomes apparent that Ki67 quantification is done by manually counting ki67+ b-cells and the total number of cxcr4+ b-cells within each pancreatic islet. However, the authors should mention of the total number of counted cells on which this quantification is based. In addition, manually counting large numbers of cells in 3D is error prone and therefore usually performed by segmentation using imaging analysis software. This should be straightforward to do for the authors as they claim to have single cell resolution.

Thank you for your comments. As suggested, we used the nuclei detection and membrane detection in the Cell module of Imaris to automatically segment and analyse the Ki67+ b-cells and CXCR4+ b-cells. Briefly, under the Cell Creation wizard, for Ki67+ b-cells, CXCR4 channel was selected as source channel for membrane-based detection and Ki67 was selected for nuclei-based detection; for CXCR4+ b-cells, CXCR4 and TO-PRO-3/DAPI channels was selected for membrane-based detection and nuclei-based detection, respectively. The final results display accurate segmentation of cells and present the total number of these b-cells. The result of Ki67+ b-cells quantification has been replaced (Fig 5G, 6D, 6H and 7J) and this segmentation method has been added in the revised manuscript. (Page 29)

5. The comments in 4 also apply to the manual counting of macrophages.

Thank you for your suggestion. As explained in the last comment, we also used the nuclei detection and membrane detection function to accurately segment cells and re-analysis the numbers of macrophages and fibroblasts. The results have been replaced in the Fig EV6. (Page 27 and 28)

Minor comment:

The Imaging set-up, acquisition and analysis section of the methods does not mention the resolution and range of the z dimension. The authors should mention the voxel size instead of the pixel size, and mention the imaged/analysed depth for the samples.

Thank you for your comments. The voxel size of the images was 3.2 mm. The imaged and analysed range of z dimension was around 22.14 mm to 53.3 mm at 0.82 mm interval size. We have added this to the revised manuscript. (Page 23)

Referee #3:

Chen et al., present an impressive imaging encyclopedia of endocrine organs, ovary, pituitary, adrenal glands, pancreas, thyroid, testis, from young and old mice as well as humans. Very interestingly, there seems to be similarities in patterns of how vessel density and number of arteries change from young to aged between mice and humans. The images performed with a wide range of antibodies in the different organs, ages and species have been deposited at Omero, an imaging repository. This is highly commendable and points to future new development in how to work with large scale images. Moreover, the authors have put a lot of effort into the presentation, which allows the reader to follow and draw conclusions even though the amount of data is close to overwhelming. Overall, this is a paper which will inspire to a wide range of exciting, hypothesis-driven research both in the authors' own laboratory and in many others.

We are very grateful to the reviewer for the positive remarks and appreciation of the work. Further, we are also grateful for the useful comments and suggestions which has helped us to substantially improve our manuscript.

Comments

1. As far as this reviewer could find, there was no access to files at Omero and so I have not been able explore the potential use for the community and in particular, the user-friendliness.

Thank you for this query. We have incorporated the link for Omero to the revised manuscript. The link is <http://homer.ox.ac.uk/webclient/?show=project-251>
UserID: 3Dendocrine
Password: ml0l6c329
(Page 29)

2. Figure 1. Please place the tile scan of each organ with the "representative" 3D image. Or at least show the organs in the same order (in all relevant figures). Alternatively, remove the 3D images (not so useful) and just to show the tile scans, which are more interesting.

Thank you for your comment. As suggested, in Fig 1, we removed the 3D images and just displayed the tile scan of each endocrine glands with the same order in all relevant figures.

3. The schematics in Figure 2 panel M is important but it is very busy as shown. Some cells are shown twice such as CD68. The two CD68 cells appear to have different relations to the medulla and transition zones - one cell has a line drawn for the young tissue but not the other. Can this be simplified/improved? Perhaps young on one side and aged on the other of the schematics. Alternatively, for each zone (medulla, transition, cortex), show all relevant cells and thereby avoid the busy lines. Perhaps the different mural cells on one side and inflammatory cells on the other? Please indicate which vessel type is depicted here. This is an important and very useful summary schematic and should be given more space. It's very interesting to see how the mural cell composition differs between the different sections.

Thank you for raising this point. We agree. As suggested, we have divided this schematic into two parts with young and aged cell-cell interactomes (Fig 1D).

4. Figures 2 and 3; show the organs in the same order as for Fig. 1. For Figure 2, please specify how arteries were identified in the different organs. For Figure 3A, the heatmap of murine organs should not be in this figure? Moreover, considering the large variability within each organ, between vessel types etc, how have the authors come to conclude that expression of a particular marker is high or low on a single cell level rather than in average? This does not make sense as shown.

Thank you for your comments. We agree. We have modified the figures and displayed the same order of endocrine glands for Fig 1-4. α -SMA was used to identify arteries. Since α -SMA could also be expressed by some stromal cells, arteries were distinguished based on their tubular shape. In addition, we also applied light-sheet imaging of cleared whole-glands with α -SMA immunostaining (Fig 3A-E).

As suggested, we have moved the heatmap of murine organs to Fig 3F. For the expression of different markers in the heatmap, actually, we have applied the average intensity analysis for each marker based on the entire region of each organ, so this qualitative analysis indicates the low to high levels of expressions in average rather than at single-cell level. Although all the images are at single-cell resolution, we agree that this heatmap is not showing single-cell data and we have clarified this in the text. Also, we have removed the term "single-cell" for this proteomics heatmap. (Page 8)

5. In Figures 4 and 5, the authors define the vasculature in the islets of Langerhans as CD31+, Emcn high while the exocrine vasculature is Emcn low. The Emcn high vasculature diminishes with age and is sensitive to VEGFR2 inhibition with Sunitinib. Although an elegant application of findings from the broad imaging endeavor, these data are not very novel but already shown by a range of different laboratories. In relation to the previous literature, it would be useful to also show expression of PLVAP as a marker of fenestrated vessels, known to be lost upon VEGFA deprivation (see for example Kamba et al. *Am J Physiol Heart Circ Physiol.* 2006 Feb;290(2):H560-76).

Thank you for your comments. As suggested, we have conducted the PLVAP staining on young and aged pancreatic islets. The results show the PLVAP expression in some but not all

the capillaries within the islets. Further, the PLVAP+ vessels declined with age (Fig EV11D), which correlated with the loss of VEGFA expression upon ageing. (Page 10)

6. In Figures 6 and 7, the authors focus on connexin43 (Gaj1) in the pancreas vasculature, as Gaj1 increases with age. Specific deletion of Gaj1 in endothelial cells leads to a range of seemingly favorable changes such as increased vessel density and proliferation of b-cells in Langerhans islets and increased insulin production. Treatment with Carbenoxolone (CBX), a drug said to decrease expression of Gaj1, mimicks the effects of genetic deletion. CBX does not have a good reputation as a drug and is known to act on a range of pathways including NMDA receptors, GABA A receptors, and calcium channels. It would probably be best to take the CBX data out; they are not needed.

Thank you for raising this point. We agree, the administration of CBX is not cell-specific and may impact other cell types; however, it successfully mimicked the results we observe in the Gja1 mutants. We have now moved the data to supplementary materials (Fig EV15). We have also removed any emphasis on this data and have added the point that it can impact on other cell types.

We have further analysed the aged Gja1 Δ EC mutants and aged littermate control mice (Fig 7j). The related contents have been modified in the revised manuscript. (Page 14 and 15)

Dear Dr Kusumbe,

Thank you for submitting your amended manuscript (EMBOJ-2020-105242R) to The EMBO Journal. Your revised study was sent back to the referees for re-evaluation, and we have received comments from referees #1 and #3, which I enclose below. These referees find that their concerns have been sufficiently addressed and they are now broadly in favour of publication. Please note that while referee #2 was at this time not able to reassess your work, we have editorially evaluated your response to the concerns raised and found them to be satisfactorily addressed.

Thus, we are pleased to inform you that your manuscript has been accepted in principle for publication in The EMBO Journal, pending some minor remaining issues related to formatting and data representation as detailed below which need to be addressed at re-submission.

Further, I will share additional changes and comments from our production team during the next days to be considered.

Please contact me at any time if you have further questions.

Thank you for giving us the chance to consider your manuscript for The EMBO Journal. I look forward to your final revision.

Again, please contact me at any time if you need any additional help.

Kind regards,

Daniel Klimmeck

Daniel Klimmeck PhD
Editor
The EMBO Journal

- >> Specify authorship contributions for S.L.T. . Add L.L. as author in our online manuscript system.
- >> Limit keywords to up to five.
- >> Limit the abstract to maximally 175 words.
- >> Re-check order of callouts in the manuscript. Fig 3A,B are called out before Fig 2C.
- >> Dataset EV legends: There are 11 movies, which need renaming to 'Movie EV1, EV2...'. All need their legends removed from the manuscript and zipped with each respective movie file.

>> The appendix is quite long now. We suggest to upload Appendix Tables S2,5,7,8 separately in excel EV tables and adjust nomenclature accordingly. Tables legends need to be removed from the manuscript and added to respective tables.

>>EV figures. There are 15 EV figures uploaded together in one PDF. Please limit to maximally five EV figures and provide them as individual high-resolution image files. The other 10 should be added to the appendix, with their legends.

>>Please enter the complete funding information for your study into our online system.

>> Free access to the image database and remove code from the data availability section.

>> Avoid textual redundancy with your 2014 study.

- a point-by-point response to the referees' comments, with a detailed description of the changes made (as a word file).

- a word file of the manuscript text.

- individual production quality figure files (one file per figure)

- a complete author checklist, which you can download from our author guidelines

(<https://www.embopress.org/page/journal/14602075/authorguide>).

- Expanded View files (replacing Supplementary Information)

Further information is available in our Guide For Authors:

The revision must be submitted online within 90 days; please click on the link below to submit the revision online before 30th Nov 2020.

Link Not Available

Referee #1:

I did not specify any points for the Authors to address in the original version of the manuscript. This revised version is of very high quality.

Referee #3:

The revision has further increased the quality of this exciting paper. I have no further comments.

The authors performed the requested editorial changes.

Dear Dr Kusumbe,

Thank you for submitting the revised version of your manuscript. I have now evaluated your amended manuscript and concluded that the remaining minor concerns have been sufficiently addressed.

Thus, I am pleased to inform you that your manuscript has been accepted for publication in the EMBO Journal.

Please note that it is EMBO Journal policy for the transcript of the editorial process (containing referee reports and your response letter) to be published as an online supplement to each paper. I would thus like to ask for your consent on keeping the additional rebuttal figures included in this file.

Also in case you might NOT want the transparent process file published at all, you will also need to inform us via email immediately. More information is available here:
http://emboj.embopress.org/about#Transparent_Process

Please note that in order to be able to start the production process, our publisher will need and contact you regarding the following forms:

- PAGE CHARGE AUTHORISATION (For Articles and Resources)

[http://onlinelibrary.wiley.com/journal/10.1002/\(ISSN\)1460-2075/homepage/tej_apc.pdf](http://onlinelibrary.wiley.com/journal/10.1002/(ISSN)1460-2075/homepage/tej_apc.pdf)

- LICENCE TO PUBLISH (for non-Open Access)

Your article cannot be published until the publisher has received the appropriate signed license agreement. Once your article has been received by Wiley for production you will receive an email from Wiley's Author Services system, which will ask you to log in and will present them with the appropriate license for completion.

- LICENCE TO PUBLISH for OPEN ACCESS papers

Authors of accepted peer-reviewed original research articles may choose to pay a fee in order for their published article to be made freely accessible to all online immediately upon publication. The EMBO Open fee is fixed at \$5,200 (+ VAT where applicable).

We offer two licenses for Open Access papers, CC-BY and CC-BY-NC-ND.

For more information on these licenses, please visit: <http://creativecommons.org/licenses/by/3.0/> and http://creativecommons.org/licenses/by-nc-nd/3.0/deed.en_US

- PAYMENT FOR OPEN ACCESS papers

You also need to complete our payment system for Open Access articles. Please follow this link and select EMBO Journal from the drop down list and then complete the payment process:

https://authorservices.wiley.com/bauthor/onlineopen_order.asp

Should you be planning a Press Release on your article, please get in contact with embojournal@wiley.com as early as possible, in order to coordinate publication and release dates.

On a different note, I would like to alert you that EMBO Press is currently developing a new format for a video-synopsis of work published with us, which essentially is a short, author-generated film explaining the core findings in hand drawings, and, as we believe, can be very useful to increase visibility of the work.

Please see the following link for a representative example:

http://embopress.org/video_EMBOJ-2014-90147

If you have any questions, please do not hesitate to call or email the Editorial Office.

Kind regards,

Daniel Klimmeck

Daniel Klimmeck, PhD
Editor
The EMBO Journal
EMBO
Postfach 1022-40
Meyerohofstrasse 1
D-69117 Heidelberg
contact@embojournal.org
Submit at: <http://emboj.msubmit.net>

YOU MUST COMPLETE ALL CELLS WITH A PINK BACKGROUND ↓
PLEASE NOTE THAT THIS CHECKLIST WILL BE PUBLISHED ALONGSIDE YOUR PAPER

Corresponding Author Name: Anjali Kusumbe

Journal Submitted to: The EMBO Journal

Manuscript Number: EMBOJ-2020-105242